# Graph Structure Learning with Interpretable Bayesian Neural Networks

**Max Wasserman**                                          *mwasser6@cs.rochester.edu*
*Department of Computer Science*
*University of Rochester*

**Gonzalo Mateos**                                          *gmateosb@ece.rochester.edu*
*Department of Electrical and Computer Engineering*
*University of Rochester*

**Reviewed on OpenReview:** *https://openreview.net/forum?id=2noXK5KBbx*

## Abstract

Graphs serve as generic tools to encode the underlying relational structure of data. Often this graph is not given, and so the task of inferring it from nodal observations becomes important. Traditional approaches formulate a convex inverse problem with a smoothness promoting objective and rely on iterative methods to obtain a solution. In supervised settings where graph labels are available, one can unroll and truncate these iterations into a deep network that is trained end-to-end. Such a network is parameter efficient and inherits inductive bias from the optimization formulation, an appealing aspect for data constrained settings in, e.g., medicine, finance, and the natural sciences. But typically such settings care equally about *uncertainty* over edge predictions, not just point estimates. Here we introduce novel iterations with *independently interpretable parameters*, i.e., parameters whose values - independent of other parameters' settings - proportionally influence characteristics of the estimated graph, such as edge sparsity. After unrolling these iterations, prior knowledge over such graph characteristics shape *prior distributions* over these independently interpretable network parameters to yield a Bayesian neural network (BNN) capable of graph structure learning (GSL) from smooth signal observations. Fast execution and parameter efficiency allow for high-fidelity posterior approximation via Markov Chain Monte Carlo (MCMC) and thus uncertainty quantification on edge predictions. Informative priors unlock modeling tools from Bayesian statistics like prior predictive checks. Synthetic and real data experiments corroborate this model's ability to provide well-calibrated estimates of uncertainty, in test cases that include unveiling economic sector modular structure from S&P500 data and recovering pairwise digit similarities from MNIST images. Overall, this framework enables GSL in modest-scale applications where uncertainty on the data structure is paramount.

## 1 Introduction

Graphs serve as a foundational paradigm in machine learning, data science, and network science for modeling complex systems, capturing intricate relationships in data that range from social networks to gene interactions; see e.g., (Kolaczyk, 2009; Hamilton, 2020). In computational biology, accurate graph structures can offer insights into gene regulatory pathways, enhancing our ability to treat diseases at the genetic level (Cai et al., 2013). In finance, the ability to recover precise financial networks can be useful for risk assessment and market stability (Marti et al., 2021). In geometric deep learning (Bronstein et al., 2017), the lack of observed graph structure underlying data often limits the use of efficient learning models such as graph neural networks (GNNs); see e.g., (Cosmo et al., 2020). Yet, despite their importance, existing methodologies for graph structure learning (GSL) from multivariate nodal data face significant limitations.

This work addresses the GSL problem where nodal observations are used to predict the completely unobserved graph structure. Traditional approaches summarize the nodal observations with a pairwise vertex (dis)similarity matrix, e.g., correlation or Euclidean distance matrices, and formulate GSL as a regularized convex inverse problem (Mateos et al., 2019; Dong et al., 2019). The primary objective encourages data fidelity, according to a chosen model linking the nodal observations and the sought latent graph, while the regularization objectives capture prior structural knowledge, such as edge sparsity or desired connectivity patterns. These *model-based* methods solve the inverse problem using optimization algorithms, which often come with convergence rate guarantees (Saboksayr & Mateos, 2021; Wang et al., 2023). However, as noted in (Pu et al., 2021) their expressiveness is constrained to graph characteristics that can be modeled via convex criteria and they may suffer complexity and scalability issues. Specifically, these model-based approaches typically necessitate an outer loop for grid-based optimization of regularization parameters (and often other algorithm parameters, e.g., a step-size), and an inner loop that, given this fully defined optimization problem, may demand thousands of iterations to converge on a solution. When nodal observations come with corresponding graph labels, recent supervised GSL approaches partially overcome these obstacles using 'algorithm unrolling' (Shrivastava et al., 2020; Pu et al., 2021; Wasserman et al., 2023). Unrollings truncate these inner-loop iterations to yield a deep network architecture that is trained end-to-end to approximate the solution to the inverse problem (Monga et al., 2021). The choice of a custom loss function, which need not be convex, plus optional architectural refinements of an already well-motivated initial network tend to improve performance on the task of interest. Truncation depth provides explicit control on the complexity of prediction, now a forward pass in the network. And the use of backpropogated gradients makes learning the regularization (and step-size) parameters, now network parameters, more scalable. Additionally, the unrolled deep networks tend to inherit appealing aspects from the original model-based formulation, namely inductive bias and low dimensionality, as their outputs are approximations to solutions of the original inverse problem.

## 1.1 Towards interpretability and uncertainty quantification: Desiderata and contributions

In composite inverse problems with multiple regularizers, understanding how each regularization parameter values affect desirable solution traits (e.g., image sharpness or number of graph edges) may be complex and often beyond the modeler's knowledge. This obscurity necessitates a naive—and consequently costly—multi-dimsioanl grid search for suitable parameter values; see e.g., the numerical study in (Dong et al., 2016, Section V-B). When the relationship between the regularization parameters and a solution characteristic is clear, we say the parameters are interpretable with respect to (w.r.t.) the output characteristic. When an output characteristic is influenced by a single parameter, independent of all others, we call this parameter *independently interpretable* w.r.t. the output characteristic. Such instances make interpretability actionable, namely they allow prior knowledge on the solution characteristic to be incorporated onto the value of the independently interpretable parameter. A key contribution of this work is: i) in recognizing optimization problems with independently interpretable parameters as ripe for the adoption of unrollings, which inherit this interpretability because they approximate inverse problem solutions; and ii) in leveraging the Bayesian framework to seamlessly incorporate such prior knowledge into the parameters of the unrolled network.

An issue unrollings often face (within GSL and beyond) is their tendency to produce layers with pre-activation outputs that are nonlinear functions of the parameters; e.g., parameter products and parameters in denominators (Monga et al., 2021). This prevents the deep network from being a *true neural network* (NN), i.e., a function composed of layers, where each layer consists of affine transformations of data or intermediate activations followed by non-linear functions applied pointwise (Bishop, 1995). To address this issue, network designers may opt for reparameterization, but at the expense of parameter interpretability and often degraded empirical performance (Monga et al., 2021; Shrivastava et al., 2020). Unrollings can be impeded by nuisance parameters - parameters not of direct interest, but which must be accounted for - like a step-size. Nuisance parameters are typically a vestige of the chosen optimization algorithm rather than being intrinsic to the problem formulation. Nuissance parameters can also undermine training stability (see Appendix A.1.1) and since they lack a clear connection to specific solution characteristics they hinder informative prior modeling. The preceding discussion motivates the need for innovative GSL techniques to produce true NN unrollings amenable to incorporation of prior knowledge on their parameters.

Providing estimates of uncertainty on an inferred graph structure is important for downstream applications, e.g. in biology, finance, and machine learning with GNNs. In general Bayesian statistics, low-dimensional models with interpretable parameters are constructed using background information on the specific problem. Interpretability allows informative prior modeling and thus access to enticing Bayesian modeling tools, e.g., prior predictive checks (Gabry et al., 2017), while the low dimensionality allows tractable high-quality posterior inference, typically via Markov Chain Monte Carlo (MCMC) sampling. For example, traditional Bayesian approaches to GSL tend to address the transductive (tied to a single graph) setting by constructing a joint model over parameters, observed data, and latent graph structure, use MCMC to draw samples, and marginalize out all but the graph samples (Butts, 2003; Crawford, 2015; Gray et al., 2020). Stepping back from the specific case of GSL, the inability to specify a sufficiently expressive model often motivates the use of Bayesian neural networks (BNNs), an alternative paradigm which typically bypasses domain-specific modeling opting instead to feed the output of a performant NN directly into the likelihood function; see e.g., (Jospin et al., 2022). Both traditional Bayesian and BNN approaches derive uncertainty estimates over predictions by marginalizing out the parameter posterior. However, BNNs present notable challenges, especially in the incorporation of prior information and posterior approximation. Due to the non-interpretability of parameters, priors are often selected for computational convenience, such as a zero-mean isotropic Gaussian, which can inadvertently bias predictions *against* prior beliefs, a phenomenon known as 'unintentionally informative priors' (Wenzel et al., 2020), thus negating a key benefit of Bayesian methods. Additionally, the parameters' high dimensionality and complex posterior geometry make high-quality posterior approximation a formidable task, often requiring significant approximations that undermine the interpretability of the results (Gal & Ghahramani, 2016; Wenzel et al., 2020; Monga et al., 2021). This highlights the need for an inductive GSL method that not only provides reliable uncertainty estimates over edge predictions but also combines the expressive power of BNNs with the traditional Bayesian approach's strengths in integrating prior knowledge, employing robust modeling tools like predictive checks, and ensuring high-fidelity posterior approximation.

**Summary of contributions.** In this paper, we introduce the first BNN for supervised GSL from smooth signal observations. This BNN produces a distribution over unseen test graphs allowing estimation of uncertainty over edge predictions. It leverages the independent intepretability of the paramters in the GSL formulation to allow informative prior modeling over the weights of the NN, itself a result of unrolling novel iterations for a model-based formulation with well-documented merits. We make the following technical contributions and offer experimental evidence to support our claims:

- In Section 3, we develop a novel optimization algorithm for GSL from smooth signals (Algorithm 1), which is step-size free and parameterized to yield independent interpretability w.r.t. edge sparsity.

- In Section 4, we unroll Algorithm 1 to produce the first strict unrolling for GSL from smooth signals, which results in a true NN. This is to be contrasted with existing supervised-learning approaches to GSL, where GLAD (Shrivastava et al., 2020) and Unrolled PDS (Pu et al., 2021) are not true NNs, and GDN (Wasserman et al., 2023) is not a strict unrolling because it resorts to gradient truncation and reparameterization. The proposed unrolled NN is used to define our BNN, dubbed 'DPG' since Algorithm 1 is a dual-based proximal gradient method (Beck & Teboulle, 2014).

- In Section 5, we introduce a methodology to integrate prior knowledge into BNN prior distributions, specifically for networks derived from unrolled optimization algorithms for inverse problems with independent interpretability. We show this approach unlocks classical Bayesian modeling tools like predictive checking, which we fruitfully apply to DPG. High-fidelity parameter posterior inference via Hamiltonian Monte Carlo (HMC) sampling enables the first instance of a model for GSL from smooth signals, capable of producing estimates of uncertainty over edge predictions.

- In Section 6, we validate DPG's ability to produce high-quality and well-calibrated uncertainty estimates from synthetic data, stock price time series (S&P500), as well as graphs learnt from images of MNIST digits. The reliability of these estimates is underscored by notable Pearson correlations between predictive uncertainty and error: 0.70 for the stock data and 0.62 for the MNIST digits. Additional experimental evidence is provided in the appendices.

## 2 Related Work

A recent body of work addresses the GSL task via algorithm unrollings under varying data assumptions. Noteworthy contributions include GLAD (Shrivastava et al., 2020), which unrolls alternating-minimization iterations for Gaussian graphical model selection, Unrolled PDS (Pu et al., 2021) which unrolls primal-dual splitting (PDS) iterations for the GSL problem from smooth signals, and GDN (Wasserman et al., 2023) which unrolls linearized proximal-gradient iterations for a network deconvolution task that posits a polynomial relationship between the observed pairwise vertex distance matrix and the latent graph. Here we deal with the GSL problem from observations of smooth signals as in (Pu et al., 2021), but the optimization algorithm we unroll is different (cf. DPG in Algorithm 1 versus PDS), more compact and devoid of step-sizes. Additionally, none of the previous GSL unrollings result in true NNs, provide estimates of uncertainty on their adjacency matrix predictions, nor leverage the interpretability of network parameters in the modeling process.

Building a probabilistic model to connect observed network data to latent graph structure has a long history in the computer and social network analysis communities; see e.g., (Coates et al., 2002; Butts, 2003; Kolaczyk, 2009). Gibbs sampling was used mostly as a means to efficiently explore the resulting network posterior rather than quantify the uncertainty the model placed on particular edges. Since then, Bayesian methods have been used with alternate network models (e.g., directed acyclic graphs specifying the structure of Bayesian networks), types of observed data (e.g., information cascades and protein-protein interactions), and posterior approximation approaches; see (Shaghaghian & Coates, 2016; Gray et al., 2020; Jiang & Kolaczyk, 2011; Williamson, 2016; Pal & Coates, 2019; Deleu et al., 2022). Such approaches are typically transductive - tying themselves to a single training graph - and require expensive *joint* inference of the model and the latent graph structure. Our approach only requires inference of the BNN model parameters and thus is naturally inductive, i.e., able to generalize to new nodes, or entirely new graphs. Some recent works build on such graph distribution modeling approaches in an approximate Bayesian manner, to incorporate the uncertainty in an observed graph for downstream tasks with GNNs (Zhang et al., 2019; Pal et al., 2020). Others forgo modeling the distribution of the observed graph but take approximate Bayesian approaches to modeling with GNNs. For instance, (Opolka & Lió, 2022) uses a deep graph convolutional Gaussian process with variational posterior approximation for link prediction, and (Sevilla & Segarra, 2023) pre-trains a score-matching GNN for use in annealed Langevin diffusion to draw approximate samples from the network posterior. All such Bayesian GNN approaches require (partial) observation of graph structure, and rely on approximate inference methods due to large dimensionality. Our DPG approach uses no observed graph structure (except for graph labels during training) and allows for high-fidelity posterior approximation. To the best of our knowledge, BNNs have so far not been used for GSL with uncertainty quantification.

More broadly, unrolling-inspired Bayesian deep networks have recently found success in uncertainty quantification for computational imaging (Barbano et al., 2020; Zhang et al., 2021; Ekmekci & Cetin, 2022). The inductive bias provided by the original iterations lead to gains in data efficiency, but still have limited parameter interpretability and high dimensionality leading to naive priors and coarse posterior approximation. This exciting line of work inspired some crucial ideas in this paper, cross-pollinating benefits to GSL and with the added value of overcoming the aforementioned Bayesian modeling and inference challenges.

## 3 Model-based Formulation and Optimization Preliminaries

Let $\mathcal{G}(\mathcal{V}, \mathcal{E}, \boldsymbol{A})$ be an undirected graph, where $\mathcal{V} = \{1, \ldots N\}$ are the vertices (or nodes), $\mathcal{E} \subseteq \mathcal{V} \times \mathcal{V}$ are the edges, and $\boldsymbol{A} \in \mathbb{R}_+^{N \times N}$ is the symmetric adjacency matrix collecting the non-negative edge weights. For $(i, j) \notin \mathcal{E}$ we have $A_{ij} = 0$. We exclude the possibility of self loops, so that $\boldsymbol{A}$ is hollow meaning $A_{ii} = 0$, for all $i \in \mathcal{V}$. For the special case of unweighted graphs that will be prominent in our models, then $\boldsymbol{A} \in \{0, 1\}^{N \times N}$.

In this paper, we consider that $\boldsymbol{A}$ is unknown and we want to estimate the latent graph structure from nodal measurements only[1]. To this end, we acquire graph signal observations $\boldsymbol{x} = [x_1, \ldots, x_N]^\top \in \mathbb{R}^N$, where $x_i$ denotes the signal value (i.e., a nodal attribute or feature) at vertex $i \in \mathcal{V}$. When $P$ such signals are available we construct matrix $\boldsymbol{X} = [\boldsymbol{x}_1, \ldots, \boldsymbol{x}_P] \in \mathbb{R}^{N \times P}$, where each row $\bar{\boldsymbol{x}}_i^\top \in \mathbb{R}^P$, $i = 1, \ldots, N$, of $\boldsymbol{X}$ represents a

---

[1]This is different to the link prediction task, where one is given measurements of edge status for a training subset of node pairs (plus, optionally, node attributes), and the transductive goal is to predict test links from the same graph(Kolaczyk, 2009)

vector of features or nodal attributes at vertex $i$. We can summarize this dataset using a pairwise vertex dissimilarity matrix, here the Euclidean distance matrix $\boldsymbol{E} \in \mathbb{R}_+^{N \times N}$, where $E_{ij} = \|\bar{\boldsymbol{x}}_i - \bar{\boldsymbol{x}}_j\|_2^2$. Assuming our data lie on a smooth manifold, we interpret $\mathcal{G}$ as a discrete representation of this manifold. When nodes $i \neq j \in \mathcal{V}$ have large edge weight $A_{ij}$, reflecting close points on the manifold, $E_{ij}$ will be small. Accordingly, smooth (w.r.t. $\mathcal{G}$) vectors in $\boldsymbol{X}$ have small total variation or Dirichlet energy (Belkin & Niyogi, 2001), namely

$$TV_{\mathcal{G}}(\boldsymbol{X}) = \frac{1}{2} \sum_{i,j} A_{ij} \|\bar{\boldsymbol{x}}_i - \bar{\boldsymbol{x}}_j\|_2^2 = \|\boldsymbol{A} \circ \boldsymbol{E}\|_{1,1}, \tag{1}$$

where $\|\boldsymbol{Z}\|_{1,1} = \sum_{i,j} |Z_{ij}|$ is the entrywise $\ell_1$-norm of matrix $\boldsymbol{Z}$ and $\circ$ is the Hadamard (entrywise) product. The prevalence of smooth network data, for instance sensor measurements (Chepuri et al., 2017), protein function annotations (Kolaczyk, 2009), and product ratings (Huang et al., 2018), justifies using a smoothness criterion for the GSL task.

### 3.1 Graph structure learning from smooth signals

Given $\boldsymbol{X}$ assumed to be smooth on $\mathcal{G}$, a popular model-based GSL approach is to minimize the Dirichlet energy in (1) w.r.t. $\boldsymbol{A}$; see e.g., (Hu et al., 2013; Dong et al., 2016; Kalofolias, 2016; Kalofolias & Perraudin, 2017). The inverse problems posed in these works can be unified under the general composite formulation

$$\boldsymbol{A}^* = \underset{\boldsymbol{A} \in \mathcal{A}}{\arg\min} \left\{ \|\boldsymbol{A} \circ \boldsymbol{E}\|_{1,1} + h(\boldsymbol{A}) \right\}, \tag{2}$$

where the feasible set is $\mathcal{A} := \{\boldsymbol{A} \in \mathbb{R}^{N \times N} : \text{diag}(\boldsymbol{A}) = \boldsymbol{0}, A_{ij} = A_{ji} \geq 0, \forall i,j \in \mathcal{V}\}$, i.e., hollow, symmetric, non-negative matrices. The regularization term $h(\boldsymbol{A})$ typically promotes desired structure on the estimated edge set (e.g., sparsity, no isolated nodes) and can be used to avoid the trivial solution $\boldsymbol{A}^* = \boldsymbol{0}$. We henceforth use $h(\boldsymbol{A}) = -\alpha \boldsymbol{1}^\top \log(\boldsymbol{A}\boldsymbol{1}) + \frac{\beta}{2}\|\boldsymbol{A}\|_F^2$ ($\alpha, \beta \geq 0$ are regularization parameters), which excludes the possibility of isolated nodes and has achieved state-of-the-art results (Kalofolias, 2016).

It is convenient to reformulate (2) in an unconstrained, yet equivalent form. We start by compactly representing variable $\boldsymbol{A}$ and data matrix $\boldsymbol{E}$ with their vectorized upper triangular parts $\boldsymbol{a}, \boldsymbol{e} \in \mathbb{R}_+^{N(N-1)/2}$, implicitly enforcing symmetry and hollowness, while also halving the problem dimension. To enforce non-negativity the indicator function $\mathbb{I}\{\boldsymbol{a} \geq 0\} = \{0 \text{ if } \boldsymbol{a} \geq 0 \text{ else } \infty\}$ is included in the objective. Finally, we substitute the nodal degrees $\boldsymbol{d} = \boldsymbol{A}\boldsymbol{1}$ with the vectorized equivalent $\boldsymbol{d} = \boldsymbol{S}\boldsymbol{a}$, where $\boldsymbol{S} \in \{0,1\}^{N \times N(N-1)/2}$ is a fixed binary matrix that maps vectorized edge weights to degrees. The resulting optimization problem is given by

$$\boldsymbol{a}^*(\boldsymbol{e}, \alpha, \beta) = \underset{\boldsymbol{a} \in \mathbb{R}^{N(N-1)/2}}{\arg\min} \left\{ 2\boldsymbol{a}^\top \boldsymbol{e} - \alpha \boldsymbol{1}^\top \log(\boldsymbol{S}\boldsymbol{a}) + \frac{\beta}{2}\|\boldsymbol{a}\|_2^2 + \mathbb{I}\{\boldsymbol{a} \geq 0\} \right\}, \tag{3}$$

which is convex and admits a unique optimal solution; see e.g., (Saboksayr & Mateos, 2021). Next, we comment on the role of the regularization parameters $\alpha, \beta$ and their interpretability properties. We then offer a brief discussion on optimization algorithms to tackle problem (3). These ingredients will be essential to build a BNN model for supervised GSL in Sections 4 and 5.

**Independent interpretability of regularization parameters.** Definition 1 formalizes the notion of independent interpretability of regularization parameters in inverse problems such as (3).

**Definition 1.** Let $\boldsymbol{x}^*(\mu_1, \ldots, \mu_n) \in \mathcal{X}$ be the solution to an inverse problem that depends on regularization parameters $\mu_1, \ldots, \mu_n$. Consider some scalar function of the solution $f : \mathcal{X} \mapsto \mathbb{R}$. In general, the value $f(\boldsymbol{x}^*)$ depends on all $\mu_1, \ldots, \mu_n$. When $f(\boldsymbol{x}^*)$ depends solely on a *single* regularization parameter $\mu_i$, then we say that $\mu_i$ is *independently interpretable* w.r.t. $f(\boldsymbol{x}^*)$.

The weights $\alpha$ and $\beta$ are not independently interpretable w.r.t. to relevant graph characteristics, frustrating straightforward interpretation of their effect on the solution $\boldsymbol{a}^*(\boldsymbol{e}, \alpha, \beta)$. Specifically, for fixed $\alpha$, increasing $\beta$ leads to denser edge patterns, as we have (quadratically) increased the relative cost of large edge weights. Indeed, the sparsest graph is obtained for $\beta = 0$. But in general, many interesting graph characteristics, e.g., sparsity, connectivity, diameter, and edge weight magnitude, are non-trivial functions of *both* $\alpha$ and $\beta$; see also (Dong et al., 2016) for a similar issue.

To facilitate *independent* control over the sparsity pattern and scale of the edge weights of recovered graphs, (Kalofolias, 2016, Prop. 2) introduced an equivalent $(\theta, \delta)$-parameterization of (3), namely

$$\boldsymbol{a}^*(\boldsymbol{e}, \alpha, \beta) = \sqrt{\frac{\alpha}{\beta}} \boldsymbol{a}^* \left( \frac{1}{\sqrt{\alpha\beta}} \boldsymbol{e}, 1, 1 \right) = \delta \boldsymbol{a}^*(\theta \boldsymbol{e}, 1, 1). \tag{4}$$

We can map from the former parameterization to the latter by first scaling $\boldsymbol{e}$ by $\theta = 1/\sqrt{\alpha\beta}$, solving (3) with $\theta \boldsymbol{e}$ using $\alpha = \beta = 1$, and finally scaling the recovered edges by the constant $\delta = \sqrt{\alpha/\beta}$; we refer the reader to (Kalofolias, 2016; Kalofolias & Perraudin, 2017) for a proof of the equivalence claim. Due to the separable structure of the right-hand-side of (4), any GSL algorithm would require a single input parameter $\theta$, and the obtained solution $\boldsymbol{a}^*(\theta \boldsymbol{e}, 1, 1)$ can then be scaled by $\delta > 0$. All in all, the sparsity level of $\boldsymbol{a}^*$ is determined solely by $\theta$, making $\theta$ independently interpretable w.r.t. sparsity. Indeed, this satisfies Definition 1 with the identifications $\boldsymbol{x}^*(\mu_1, \ldots, \mu_n) \leftarrow \delta \boldsymbol{a}^*(\theta \boldsymbol{e}, 1, 1)$, $\mathcal{X} \leftarrow \mathbb{R}_+^{N(N-1)/2}$, $\mu_i \leftarrow \theta$, and $f(\boldsymbol{x}^*) \leftarrow \|\delta \boldsymbol{a}^*(\theta \boldsymbol{e}, 1, 1)\|_0$, where $\|\cdot\|_0$ counts the number of non-zero elements of its vector argument. Moreover, $\delta$ is interpretable w.r.t. edge weight magnitude, but not independently so, as larger $\theta$ produces smaller weights [see Figure 3 (bottom-left)].

## 3.2 Optimization algorithms

Problem (3) has a favorable structure that has been well documented, and several efficient optimization algorithms were proposed to obtain a solution $\boldsymbol{a}^*(\boldsymbol{e}, \alpha, \beta)$ with $\mathcal{O}(N^2)$ complexity per iteration. Specifically, a forward-backward-forward PDS algorithm was first proposed in (Kalofolias, 2016). PDS introduces a step-size parameter which must be tuned to yield satisfactory empirical convergence properties, thus increasing the overall computational burden. We find that effective step-size values tend to lie on a narrow interval beyond which PDS exhibits divergent behavior, further frustrating tuning; see Appendix A.1.1 for a supporting discussion. GSL algorithms based on the alternating-directions method of multipliers (Wang et al., 2021) or majorization-minimization (Fatima et al., 2022) have been developed as well. Recently, (Saboksayr & Mateos, 2021) introduced a fast dual proximal gradient (FDPG) algorithm to solve (3), which is devoid of step-size parameters and – different from all three previous approaches – it comes with global convergence rate guarantees. For this problem, the strongest convergence results to date are in (Wang et al., 2023).

Our starting point in this work is the FDPG optimization framework, but different from (Saboksayr & Mateos, 2021) we: (i) develop a solver for the $(\theta, \delta)$-parameterization of (3); and (ii) turn-off the Nesterov-type acceleration from the proximal-gradient iterations used to solve the dual problem of (3). This yields a dual proximal gradient (DPG) method, tabulated under Algorithm 1. In a nutshell, during iterations $k = 1, 2, \ldots$ Algorithm 1 updates the vectorized adjacency matrix estimate $\boldsymbol{a}_k \in \mathbb{R}_+^{N(N-1)/2}$, an auxiliary vector of nodal degrees $\boldsymbol{d}_k \in \mathbb{R}_+^N$, as well as dual variables $\boldsymbol{\lambda}_k \in \mathbb{R}^N$ used to enforce the variable splitting constraint $\boldsymbol{d} = \boldsymbol{S}\boldsymbol{a}$. A naive DPG implementation incurs $\mathcal{O}(N^2)$ computational and memory complexities, and we note all nonlinear operations involved (i.e., $\text{ReLU}(\cdot) = \max(0, \cdot)$, $(\cdot)^2$, and $\sqrt{(\cdot)}$) are pointwise on their vector arguments. As a result of the design choices (i)-(ii), the DPG algorithm requires the fewest operations per iteration and the fewest number of parameters among existing solvers of (3), and is devoid of any uninterpretable nuisance parameters, e.g., step-sizes. FDPG was only considered on the original $(\alpha, \beta)$-parameterization of (3); by instead opting for DPG iterations to solve the $(\theta, \delta)$-parameterization of (3), we reveal independent interpretability of $\theta$ w.r.t. sparsity of the optimal graphs.

Next, we will unroll Algorithm 1 to produce a GSL NN which inherits its advantages - namely simple, efficient, minimally parameterized layers, with independent interpretability - forming the backbone of our BNN.

## 4 Graph Structure Learning from Smooth Signals with Bayesian Neural Networks

So far we have described a model-based approach to (point) estimation of graphs from smooth signals. In this work, we assume a labeled training dataset is available. We aim to construct a BNN model to produce uncertainty estimates on graph predictions for unseen test data.

**Our BNN approach for GSL in a nutshell.** Here, we restrict ourselves to binary graphs $\boldsymbol{a} \in \{0, 1\}^{N(N-1)/2}$; weighted graphs only require a change to the ensuing likelihood function. We denote all training data as $\mathcal{T} = \{\mathcal{T}_e, \mathcal{T}_a\} = \{\boldsymbol{e}^{(t)}, \boldsymbol{a}^{(t)}\}_{t=1}^T$, an unseen test sample as $(\tilde{\boldsymbol{e}}, \tilde{\boldsymbol{a}})$, and the collection of

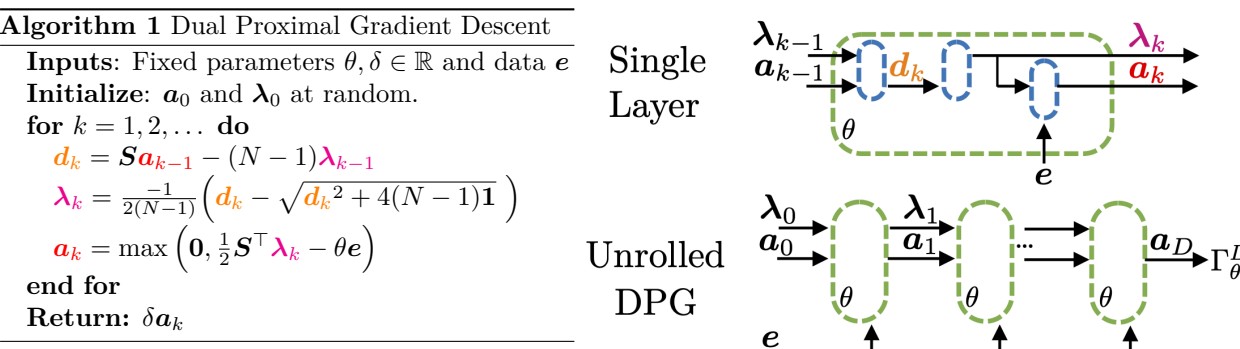

**Algorithm 1** Dual Proximal Gradient Descent

**Inputs**: Fixed parameters $\theta, \delta \in \mathbb{R}$ and data $\boldsymbol{e}$
**Initialize**: $\boldsymbol{a}_0$ and $\boldsymbol{\lambda}_0$ at random.
**for** $k = 1, 2, \ldots$ **do**
$\quad \boldsymbol{d}_k = \boldsymbol{S}\boldsymbol{a}_{k-1} - (N-1)\boldsymbol{\lambda}_{k-1}$
$\quad \boldsymbol{\lambda}_k = \frac{-1}{2(N-1)}\left( \boldsymbol{d}_k - \sqrt{\boldsymbol{d}_k^2 + 4(N-1)\mathbf{1}} \right)$
$\quad \boldsymbol{a}_k = \max\left( \mathbf{0}, \frac{1}{2}\boldsymbol{S}^\top \boldsymbol{\lambda}_k - \theta\boldsymbol{e} \right)$
**end for**
**Return**: $\delta\boldsymbol{a}_k$

Figure 1: *Left:* The dual proximal gradient (DPG) algorithm to solve the GSL problem (3). *Right:* Unrolling and truncating Algorithm 1 after $D$ iterations produces Unrolled DPG.

all parameters of our yet to be defined BNN model as $\boldsymbol{\Theta}$. Following the principles of BNNs, our goal is to construct a posterior predictive distribution $p(\tilde{\boldsymbol{a}} \mid \tilde{\boldsymbol{e}}, \mathcal{T})$ by marginalizing out model parameters $\boldsymbol{\Theta}$, i.e.,

$$p(\tilde{\boldsymbol{a}} \mid \tilde{\boldsymbol{e}}, \mathcal{T}) = \int p(\tilde{\boldsymbol{a}} \mid \tilde{\boldsymbol{e}}, \boldsymbol{\Theta}) \cdot p(\boldsymbol{\Theta} \mid \mathcal{T})\, \mathrm{d}\boldsymbol{\Theta} \approx \frac{1}{M}\sum_{m=1}^{M} p(\tilde{\boldsymbol{a}} \mid \tilde{\boldsymbol{e}}, \boldsymbol{\Theta}^{(m)}), \tag{5}$$

where we use $M$ Monte Carlo draws from the posterior $\boldsymbol{\Theta}^{(m)} \sim p(\boldsymbol{\Theta} \mid \mathcal{T})$ to approximate the intractable integral. We can then designate our edge-wise point and uncertainty estimates as the first two moments of the posterior predictive marginals $p(\tilde{a}_i \mid \tilde{\boldsymbol{e}}, \mathcal{T})$, respectively, for each candidate edge indexed by $i \in \mathcal{V} \times \mathcal{V}$.

**Roadmap.** In Section 4.1, we first discuss the development of a GSL NN which takes in the vectorized Euclidean distance matrix $\boldsymbol{e}$ of nodal observations $\boldsymbol{X}$ and assigns a probability to all possible edges. We do so by unrolling Algorithm 1, producing an GSL NN with an independently interpretable parameter w.r.t. sparsity of graph outputs. Treating the model parameters of this unrolled NN as stochastic, choosing a likelihood function (Section 4.2), and setting appropriate priors (Section 5), we produce a BNN. For inference we use HMC (Section 4.3), an unusual choice in BNNs made viable by the low dimensionality and fast execution of our NN, stemming from its origins as an unrolling. Pushing the test inputs through each such GSL NN and averaging the resulting predictive distributions provides provides the approximate (5) posterior predictive, from which we derive edge-wise point and uncertainty estimates as elaborated in Section 4.4.

## 4.1 Algorithm unrolling: Iterative optimization as a neural network blueprint

Unrollings (also known as deep unfoldings) use iterative algorithms as templates of neural architectures that are trained to approximate solutions of optimization problems. Iterations are truncated and mapped to NN layers, while optimization (e.g., regularization and step-size) parameters are turned into learnable weights. Originating from convergent iterative procedures, unrolled NNs inherit many desirable properties; namely, a low number of parameters, fast layer-wise execution, and favorable generalization in low-data environments (Monga et al., 2021). Typical architectural design choices include whether to replace intermediate operators with more expressive (possibly pre-trained) learnable modules - making them no longer 'strict' unrollings - and whether a common set of parameters should be used in all layers, or, to decouple these parameters across layers. Both decisions are context dependent and often driven by the amount of data available as well as complexity considerations. Deviating from strict unrollings with shared parameters can naturally lead to gains in expressive power, but often at the cost of inductive bias and training stability.

**Unrolling DPG iterations**. Pioneered by (Gregor & LeCun, 2010) to efficiently learn sparse codes from data, algorithm unrolling ideas are recently gaining traction for GSL as well. Starting from the formulation in Section 3.1, (Pu et al., 2021) unrolls a PDS solver of the $(\alpha, \beta)$-parameterization (3). Unrolled PDS has no independently interpretable parameters, is not a true NN (pre-activation outputs are nonlinear functions of the parameters), and includes a nuisance step-size parameter – arguably a shortcoming as gradient-based

learning can easily produce large-enough weight values for divergent behavior, and thus NaN's. Attempting to unroll PDS iterates for the $(\theta, \delta)$-parameterization of (3) would not fix these practical problems.

We instead advocate unrolling the DPG iterations (Algorithm 1) developed to solve the $(\theta, \delta)$-parameterization of (3). This way, we obtain a *true NN without nuisance step-size parameters* - avoiding the aforementioned issues and reducing the parameter count by a third - while inheriting independently interpretable parameter $\theta$ (w.r.t. sparsity of graph outputs). Incidentally, layers in Unrolled DPG [depicted in Figure 1 (right)] are markedly simpler and require fewer operations than Unrolled PDS. We denote the output of a $D$-layer unrolling of Algorithm 1 as $\delta\Gamma_\theta^D$. As we would like probabilities over candidate edges, we subtract a learnable mean shift $b$ and drive the output through a sigmoid $\sigma(\cdot)$, producing our desired GSL NN output

$$\hat{\boldsymbol{p}} = \sigma(\delta\Gamma_\theta^D(\boldsymbol{e}) - b\mathbf{1}) \in (0,1)^{N(N-1)/2}, \tag{6}$$

with parameters $\boldsymbol{\Theta} = \{\theta, \delta, b\}$. To see that Unrolled DPG is a true NN, note that $\theta$ is only involved in a linear function $\theta\boldsymbol{e}$ of the input data. Likewise, $\delta$ and $b$ are only involved in an affine mapping of the activations $\Gamma_\theta^D(\boldsymbol{e})$. All non-linear operations (specifically squaring, square root, and max) are pointwise functions of intermediate activations. Going back to the design considerations mentioned at the beginning of this section, here we keep the unrolling strict and share parameters across layers to retain independent interpretability of $\theta$, minimize parameter count, and simplify upcoming Bayesian inference. Tradeoffs arising with model expansion using multiple input and output channels per layer are discussed in Section 6.1.

All in all, unrolled DPG is the first true NN for GSL from smooth signals, and the first strict unrolling for GSL which produces a true NN. For the various reasons laid out in the preceding discussion, Unrolled DPG is of independent interest as a new model for point estimation of graph structure in a supervised setting. As we show next, it will be an integral component of the stochastic model used to construct a BNN to facilitate uncertainty quantification for adjacency matrix predictions.

## 4.2 Stochastic model

Here we specify a stochastic model for the random variables of interest, namely the binary adjacency matrix $\boldsymbol{a}$, nodal data entering via the Euclidean distance matrix $\boldsymbol{e}$, and the BNN weights $\boldsymbol{\Theta}$. The Bayesian posterior from which we wish to sample satisfies $p(\boldsymbol{\Theta} \mid \mathcal{T}) \propto p(\mathcal{T}_a \mid \mathcal{T}_e, \boldsymbol{\Theta})p(\boldsymbol{\Theta})$, and a first step in BNN design is to specify the likelihood $p(\boldsymbol{a} \mid \boldsymbol{e}, \boldsymbol{\Theta})$ and the prior $p(\boldsymbol{\Theta})$. An i.i.d. assumption on the training data $\mathcal{T}$ allows the likelihood to factorize over samples $p(\mathcal{T}_a \mid \mathcal{T}_e, \boldsymbol{\Theta}) = \prod_{t=1}^T p(\boldsymbol{a}^{(t)} \mid \boldsymbol{e}^{(t)}, \boldsymbol{\Theta})$. Moreover, assuming edges within a graph sample $\boldsymbol{a}^{(t)}$ are mutually conditionally independent given parameters $\boldsymbol{\Theta}$ leads to further likelihood factorization as $p(\mathcal{T}_a \mid \mathcal{T}_e, \boldsymbol{\Theta}) = \prod_{t=1}^T \prod_{i=1}^{N(N-1)/2} p(a_i^{(t)} \mid \boldsymbol{e}^{(t)}, \boldsymbol{\Theta})$. We model $a_i \mid \boldsymbol{e}, \boldsymbol{\Theta} \sim \text{Bernoulli}(\hat{p}_i)$, where the success (or edge $i \in \mathcal{V} \times \mathcal{V}$ presence) probability is given by the output of the Unrolled DPG network, i.e., $\hat{p}_i = \sigma\left(\delta[\Gamma_\theta^D(\boldsymbol{e})]_i - b\right)$ as in (6). Putting all the pieces together, the final expression for the likelihood is

$$p(\mathcal{T}_a \mid \mathcal{T}_e, \boldsymbol{\Theta}) = \prod_{t=1}^T \prod_{i=1}^{N(N-1)/2} (\hat{p}_i^{(t)})^{a_i^{(t)}} (1 - \hat{p}_i^{(t)})^{1-a_i^{(t)}}. \tag{7}$$

We reiterate that, crucially, the Unrolled DPG outputs enter the stochastic model via the likelihood, as the means of the edge distributions. The specification of the prior $p(\boldsymbol{\Theta})$ will be addressed in Section 5.

## 4.3 Inference

We aim to generate $M$ Monte Carlo draws from the posterior $\boldsymbol{\Theta}^{(m)} \sim p(\boldsymbol{\Theta} \mid \mathcal{T}) \propto p(\mathcal{T}_a \mid \mathcal{T}_e, \boldsymbol{\Theta})p(\boldsymbol{\Theta})$. In modern BNNs based on overparameterized deep networks, high dimensionality and parameter unidentifiability are prevalent. This translates to highly multi-modal posteriors which make sampling challenging (Izmailov et al., 2021). Even when posterior geometries are more benign, the computational and memory requirements for evaluating the network and its gradients render the generation of Monte Carlo posterior samples impractical. As a typical workaround one can resort to posterior approximation using inexpensive mini-batch methods such as mean-field variational inference or stochastic-gradient MCMC (Izmailov et al., 2021). In contrast, we

now argue that generation of high-fidelity Monte Carlo posterior samples using HMC is uniquely compatible with our proposed BNN; for implementation details see Section 6.1.

For starters, Unrolled DPG's repetitive layers simplify model construction and allow straightforward compilation in automatic differentiation software, significantly boosting runtime efficiency. The low parameter count of Unrolled DPG facilitates the use of forward-mode auto-differentiation; this eliminates the need to store intermediate activations and perform a backward pass, thus it ensures memory usage does not increase with model depth $D$. Moreover, as Unrolled DPG can effectively approximate inverse problem solutions with relatively shallow depths (see Section 6), selecting a smaller $D$ significantly reduces runtime complexity. For small- to moderately-sized graphs and datasets, our computational and memory requirements for network and gradient evaluation are low; see Section 6 for GSL problem instances where inference is attainable in $< 2$ minutes on a M2 MacBook laptop. The limited parameter count also aids in sampling efficiency as we avoid the curse of dimensionality that complicates sampling in higher-dimensional models. Finally, the ability to place informative priors over independently interpretable parameters (as we do in Section 5) is known to smoothen the posterior geometry, especially in data scarce regimes where the likelihood and prior are of comparable magnitude (Gelman et al., 1995).

### 4.4 Prediction

By conditioning on data $\mathcal{T}$ and integrating out model parameters $\boldsymbol{\Theta}$, we obtain a predictive distribution on unseen graph adjacency matrices $\tilde{\boldsymbol{a}}$ given nodal signals $\tilde{\boldsymbol{e}}$. As the integral is intractable, we use the $M$ posterior samples obtained via HMC in the inference stage to approximate the predictive distribution $p(\tilde{\boldsymbol{a}} \mid \tilde{\boldsymbol{e}}, \mathcal{T})$. This approximation process is summarized in (5). .

Importantly, we can generate samples from the posterior predictive by randomly drawing from the sampling distribution with each parameter sample plugged in, i.e., $\tilde{\boldsymbol{a}}^{(m)} \sim p(\tilde{\boldsymbol{a}} \mid \tilde{\boldsymbol{e}}, \boldsymbol{\Theta}^{(m)})$. Given the form of our BNN's stochastic model as introduced in Section 4.2, such a draw reduces to sampling a Bernoulli distribution for each possible edge. Randomly drawing from the sampling distribution is critical as it accounts for both forms of uncertainty in posterior predictive quantities, namely, sampling uncertainty and estimation uncertainty (Gelman et al., 1995). Using these posterior predictive samples, we can approximate the mean ('pred. mean') and standard deviation ('pred. stdv.') of the edge-wise marginals of the posterior predictive as

$$\mathbb{E}[\tilde{a}_i \mid \tilde{\boldsymbol{e}}, \mathcal{T}] \approx \frac{1}{M} \sum_{m=1}^{M} \tilde{a}_i^{(m)} \quad \text{and} \quad \mathrm{Var}[\tilde{a}_i \mid \tilde{\boldsymbol{e}}, \mathcal{T}]^{\frac{1}{2}} \approx \left[ \frac{1}{M-1} \sum_{m=1}^{M} (\tilde{a}_i^{(m)} - \mathbb{E}[\tilde{a}_i \mid \tilde{\boldsymbol{e}}, \mathcal{T}])^2 \right]^{\frac{1}{2}}. \tag{8}$$

Naturally, $\mathbb{E}[\tilde{a}_i \mid \tilde{\boldsymbol{e}}, \mathcal{T}]$ is a Bayesian point estimate for $\tilde{a}_i$, while $\mathrm{Var}[\tilde{a}_i \mid \tilde{\boldsymbol{e}}, \mathcal{T}]^{\frac{1}{2}}$ offers a measure of uncertainty in such prediction of edge $i$.

## 5 Bayesian Modeling of Unrolling-Based BNNs with Independent Interpretability

In this section, we present a method for Bayesian modeling for BNNs which use a network produced by unrolling an optimization algorithm to solve an inverse problem with independent interpretability. We instantiate these ideas on the GSL problem (3) - where solutions are undirected graphs - and use prior knowledge over the sparsity of such graphs to shape prior distributions over the corresponding independently interpretable parameter $\theta$ in Unrolled DPG introduced in Section 4.1.

### 5.1 Prior modeling

For Bayesian models, priors over unobserved quantities play two main roles: encoding information germane to the problem being analyzed and aiding the ensuing Bayesian inferences. Despite the name, a prior can in general only be interpreted in the context of the likelihood with which it will be paired. There are many types of priors, and we here we list the common ones in order of degree in which it is intended to affect the information in the likelihood: non-informative, reference, structural, regularizing, weakly informative, and strongly informative (Gelman et al., 2017). The more domain specific information present, the further to the

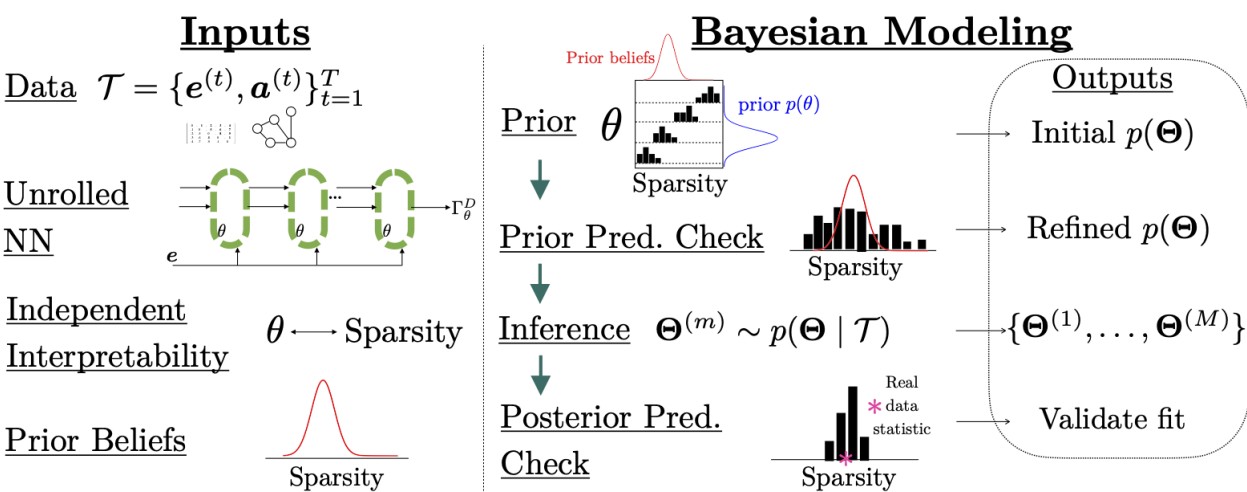

Figure 2: Bayesian workflow with independent interpretability. Inputs: A labeled data set $\mathcal{T}$, an inverse problem with independently interpretable parameter $\theta$ w.r.t. some characteristic of the solution, prior beliefs over this solution characteristic, and an unrolled NN which approximate solutions to the inverse problem. Bayesian Modeling: We use independent interpretability of $\theta$ to convert prior beliefs on solution characteristics to a prior distribution on the independently interpretable parameter. We use prior predictive checks to ensure priors generate data sets which encompass all plausible values of the solution characteristic, while still preferentially generating data sets which we believe are more likely apriori. If not, we can leverage independent interpretability to refine the prior. We then sample from the posterior, and use posterior predictive checks to provide a subjective validation of model fit.

end of this list we may be able to move, which often results in better behaved posterior geometries, and thus more stable Bayesian inference. Exactly how this domain specific information is encoded in the prior depends on both the problem and the specific chosen model.

**Informative priors with independent interpretability**. In the context of the GSL problem dealt with here, independent interpretability makes setting a strongly informative prior over DPG's independently interpretable parameter $\theta$ feasible. To this end, one first discretizes $\theta$ over a few orders of magnitude. For each $\theta$ value in the grid, we run Algorithm 1 on a subset of observed inputs $\mathcal{T}_e$ and record the relevant data characteristic (edge sparsity) of each recovered solution. We then plot the histogram of such data characteristic and observe their agreement with prior beliefs. Finally, one chooses an appropriate parametric distribution $p(\theta)$ which approximates this relative level of agreement across the range of $\theta$ values. We can use the recovered solutions at a priori probabale $\theta$ values to set weakly informative priors over the remaining model parameters. In our setting, that involves the scale of the edge weights of the recovered solutions. In Section 5.2 we discuss evaluation of this initial $p(\Theta)$ in terms of the predictions it makes. For a visualization of this prior modeling workflow, see Figure 2. A numerical example is now presented to ground this methodology.

**Example (Informative prior modeling of DPG parameters)**. In this example that continues in Section 5.2, we use a data set $\mathcal{T}$ of $T = 50$ random geometric graphs ($N = 20$ nodes, connectivity of $\frac{1}{3}$), denoted $\mathrm{RG}_{\frac{1}{3}}$, and their corresponding analytic Euclidean distance matrices as inputs defined in Section 6. See Appendix A.4 for similar analysis on other random graph distributions. Prior modeling and upcoming prior predictive checks require only the distance matrices $e$, and we use only 5 such distance matrices from $\mathcal{T}$.

Suppose we have prior beliefs on the sparsity related characteristics of recovered graphs, namely sparsity itself ($= 1 -$ edge density) or the number of connected components. For example, suppose we believe recovered graphs should have edge densities around $[.05, .5]$ and $\leq 5$ connected components. Since Unrolled DPG approximates solutions to (3) - and $\theta$ determines such sparsity related characteristics independently of other optimization parameters - we can simply run Algorithm 1 to convergence using the 5 inputs on a set of discretized $\theta$ values across several orders of magnitude, as demonstrated in Figure 3 (top-left). We observe $\theta \in [10^{-1}, 10^1]$ produces solutions with sparsity characteristics consistent with our prior beliefs. Choosing $\theta \sim$

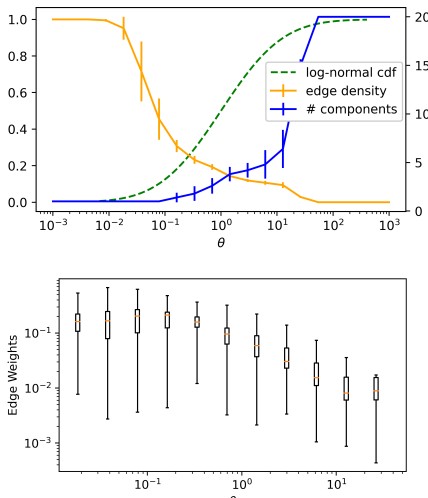
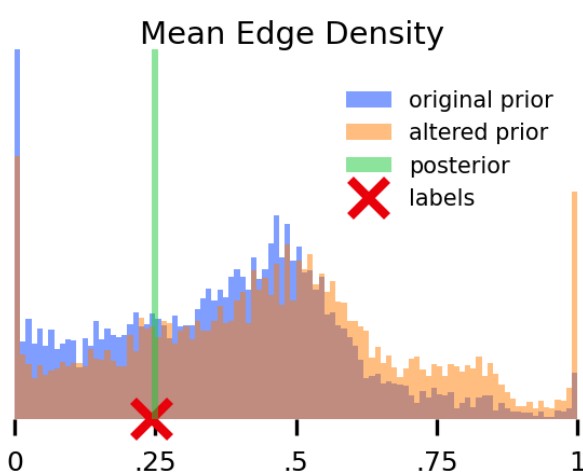

Figure 3: *Left:* Prior modeling with independently interpretable parameter $\theta$. We run Algorithm 1 to convergence over discretized $\theta$. *Top*: Larger $\theta$ produces sparser graphs. *Bottom*: Larger $\theta$ produces smaller edge weight magnitudes. *Right:* Predictive Checks. The original prior generates very few data sets with densities of $\approx .9$, a value we feel is plausible. We can use the independent interpretability of $\theta$ w.r.t. edge density to alter its prior accordingly. A prior predictive check with this altered prior now encompasses these plausible data sets. The posterior predictive check ensures the replicated data sets - now sampled after conditioning on the training data - have similar edge densities to the observed training labels. Indeed, these edge densities, denoted as 'posterior', are tightly distributed around the average edge density of the labels.

Lognormal$(0, 4)$ concentrates approximately $75\%$ probability mass on interval $[10^{-1}, 10^1]$ while still covering a broader range of values to accommodate uncertainty stemming from sampling variance (small data set) and approximation error (truncated iterations). Priors are placed over the non-independently interpretable parameters $\delta$ and $b$ with the purpose of guarding against implausible values and stabilizing posterior sampling; such priors are sometimes called 'weakly informative' (Gelman et al., 1995). Because $\delta\Gamma_\theta^D(e) - b$ will be driven through a sigmoid, we aim for both terms to be of comparable magnitude. Since our graphs are binary, $\delta\Gamma_\theta^D(e)$ can reasonably be assumed to be $\approx 1$. Utilizing Algorithm 1 solutions obtained over the $\theta$ grid, we examine the edge weight distribution (prior to scaling by $\delta$) in Figure 3 (bottom-left) to inform $p(\delta)$. Setting $\delta \sim$ Lognormal$(2, 2)$ brings scaled edge weights close to 1 for high (prior) probability values of $\theta$. Similarly, setting $b \sim$ Lognormal$(1, 2)$ ensures $b$ is also approximately 1. Both distributions assign probability mass across several orders of magnitude, both above and below 1, reflecting the breadth of our uncertainty concerning their specific value ranges. Next, we employ prior predictive checks to validate that - with the established priors - our model generates graphs that align with our prior beliefs.

## 5.2 Predictive checking

In Bayesian statistics, predictive checks evaluate a model's efficacy in capturing data characteristics like means, standard deviations, and quantiles (Gelman et al., 1995). Predictive checks leverage the existence of a model of the joint distribution $p(\mathcal{T}, \boldsymbol{\Theta})$ between data and parameters. This model allows us to draw samples from the data marginal - called *replicated* data - by drawing samples from the joint and simply dropping the parameter samples. Prior predictive checks draw replicated data from the joint before conditioning on data $\mathcal{T}$, while posterior predictive checks do so after. In both cases, we apply the chosen statistic capturing our data characteristic of interest to the replicated data, producing a histogram. The goal in prior predictive checking is to shape priors that encompass all *plausible* data sets, while still guiding the model towards data sets we deem more *likely* apriori. Thus prior predictive checking often consists of visual inspection of the histogram to ensure this holds, and adjusting the location and/or scale of prior distributions until it does. The goal in posterior predictive checking is to subjectively validate model fit; here we compare the statistic

on the replicated data (a histogram) to the same statistic applied to the actual data $\mathcal{T}$ (a single scalar). A well-fit model should have a histogram tightly concentrated around the real data statistic.

In typical BNNs, the lack of parameter interpretability prevents effective use of predictive checking. (Wenzel et al., 2020) highlights this issue by demonstrating that an isotropic Gaussian prior on NN weights (ResNet-20) fails to generate data consistent with prior expectations in image classification tasks. But the lack of parameter interpretability prevents the use of these findings to modify the prior for improved alignment. Below, we instantiate predictive checking for DPG, with average edge density as our test statistic.

**Example (Prior predictive check over DPG parameters).** To perform a prior predictive check we use the 5 inputs from $\mathcal{T}_e$ and draw $10^4$ replicated data sets $\boldsymbol{a}^{\text{rep}}$. Figure 3 (right) shows the histogram of average edge densities of replicated data sets under the prior settings laid out in Section 5.1, which we call the 'original prior'. We observe that such a prior succeeds in producing data sets that coincide with those we deem more likely apriori, but falls short of encompassing all plausible data sets, namely those with edge densities of $\approx 0.9$. To address this, we lean on independent interpretability of $\theta$ w.r.t. the sparsity of graph solutions: decreasing the location of $p(\theta)$ will increase edge densities (lower sparsity) of graph solutions, and so we lower $p(\theta)$'s location parameter from 0 to $-\frac{1}{2}$, with all other parameters of the prior distribution remaining the same. We call this new prior the 'altered prior'. Repeating the prior predictive check with this 'altered prior' we now observe $\approx 12\%$ of replicated data sets with edge densities in $[.75, 1]$, as opposed to $4\%$ before, and we confirm through visual inspection that data sets with all possible edge densities are indeed produced. The 'altered prior' thus encompasses all plausible data sets while still preferentially generating data sets we feel are more likely apriori.

**Example (Posterior predictive check over DPG parameters).** Now, we repeat this procedure, but draw replicated data sets from the joint after conditioning on $\mathcal{T}$. We perform inference drawing $M = 10^4$ posterior samples; for each posterior sample we draw a single replicated data set. In Figure 3 (right) we plot the histogram of the average edge densities of these replicated data sets, denoted 'posterior', and compare against the average edge density of the graph labels in $\mathcal{T}$, denoted 'labels'. Because all outcomes are tightly distributed around the mean edge density of the real data, we can have confidence the model parameters have fit appropriately.

# 6 Experiments

We have introduced DPG, the first BNN for GSL from smooth signals, which is capable of providing estimates of uncertainty of its edge predictions. We showed DPG can effectively incorporate prior information into the prior distribution over its parameters. In this section, we evaluate DPG across synthetic and real datasets, and introduce other baseline models for comparison, including a more expressive variant of DPG.

## 6.1 Models, metrics, and experimental details

**Strict unrollings: DPG and PDS.** In the following experiments, the prior used for the DPG's parameters $\{\theta, \delta, b\}$ is the 'altered prior' as developed in Section 5: $\theta \sim \text{Lognormal}(-1/2, 4)$, $\delta \sim \text{Lognormal}(2, 2)$, and $b \sim \text{Lognormal}(1, 2)$. We similarly refer to the BNN with Unrolled PDS as its NN model - and no change to the stochastic model - as PDS. As none of PDS's parameters $\{\alpha, \beta, \gamma, b\}$ have independent interpretability, we cannot use the prior modeling techniques outlined in Section 5. Attempting HMC on PDS produces significant number of divergent simulated Hamiltonian trajectories, a phenomenon known to be caused by high posterior curvature (Betancourt, 2016). This makes HMC run slowly, produce poorly mixed chains, and ultimately non-performant PDS models. Indeed, we find that values of step-size $\gamma$ which produce divergent behavior (very low likelihood) are close those which are most performant (very high likelihood), indicating high-curvature in the likelihood function, and thus the posterior; see Figure 8 (left). Because $\gamma$ is not interpretable, it is unclear how to shape $p(\gamma)$ to reduce posterior curvature without first running a discrete search. Indeed, to find a performant PDS model amenable to efficient HMC, we first run such a discrete search, fix $\gamma$ to a found performant value of 0.1, and then set priors $\alpha, \beta \sim \text{LogNormal}(0, 10)$ and $b \sim \mathcal{N}(0, 10^3)$. Both model-based algorithms in (Wang et al., 2023) and FDPG Saboksayr & Mateos (2021) come with faster convergence rate guarantees than DPG in Algorithm 1 - a characteristic found not to be

critical in the unrolled setting (Monga et al., 2021). They involve complex operations and have not been unrolled in prior work, unlike PDS. Specifically, (Wang et al., 2023) use 6 parameters, including 2 step-sizes, complicating stable unrolling and Bayesian inference, and so we exclude these methods from our comparisons.

**Model expansion via MIMO and partial stochasticity: DPG-MIMO and DPG-MIMO-E.** Here, we expand the Unrolled DPG NN for more expressive power by making each layer multi-input multi-output (MIMO). Each MIMO layer maps inputs $\{\boldsymbol{\lambda}_{k-1} \in \mathbb{R}^{N \times C}, \mathbf{a}_{k-1} \in \mathbb{R}^{N(N-1)/2 \times C}\}$ to outputs $\{\boldsymbol{\lambda}_k \in \mathbb{R}^{N \times C}, \mathbf{a}_k \in \mathbb{R}^{N(N-1)/2 \times C}\}$, where $C$ is the fixed number of input and output channels across layers. As before, layers share parameters, now $\boldsymbol{\theta} \in \mathbb{R}^{C \times C}$. In the $k$-th layer, parameter $\theta_{ji} \in \mathbb{R}$ is used to process $\boldsymbol{\lambda}_{k-1}[:,i]$ and $\mathbf{a}_{k-1}[:,i]$ as in a regular DPG iteration. Doing so for $i \in \{1, \dots, C\}$ and averaging the refined $\boldsymbol{\lambda}$'s and $\mathbf{a}$'s produces $\boldsymbol{\lambda}_k[:,j]$ and $\mathbf{a}_k[:,j]$, respectively. We use parameters $\boldsymbol{\delta} \in \mathbb{R}^C$ and $b \in \mathbb{R}$ to produce edge probabilities $\sigma(\frac{1}{C}\boldsymbol{a}_D\boldsymbol{\delta} - b\mathbf{1})$. The priors are unchanged from the 'altered prior' developed in the non-MIMO setting from Section 5. The increased parameterization and complexity of operations make the posterior geometry significantly more complex, such that we could not find a setup which produced well-mixed posterior sampling via HMC. Instead, we resort to Maximum a Posteriori (MAP) estimation, equivalent to maximizing the posterior with fixed observed data, using full gradient descent. We call this non-stochastic setup DPG-MIMO.

A common approach to overcome the intractable posterior inference in BNNs is partial stochasticity, where we learn point estimates of a subset of the parameters and distributions over the rest. Recent work has shown partial stochasticity of a BNN can produce similarly useful posterior predictive distributions, even outperforming fully stochastic networks in prediction in some setting (Sharma et al., 2022). Here, we 'decapitate' the MAP trained DPG-MIMO by discarding $\boldsymbol{\delta}_{\text{MAP}}$ and $b_{\text{MAP}}$; instead we feed each of the $C$ output channels $\boldsymbol{a}_D[:,j]$ of the base MAP DPG-MIMO (only parameterized by $\boldsymbol{\theta}_{\text{MAP}}$) into a new depth 20 stochastic single channel DPG $\delta_j \Gamma_{\theta_j}^{20}(\boldsymbol{a}_D[:,j])$. We average their output, shift by new stochastic $b$, and drive through a sigmoid: $\sigma\left(\frac{1}{C}\sum_{j=1}^C \delta_j \Gamma_{\theta_j}^{20}(\boldsymbol{a}_D[:,j])) - b\mathbf{1}\right)$. We can now run inference on these stochastic head parameters $\{\theta_1, \delta_1, \dots, \theta_C, \delta_C, b\}$ keeping $\boldsymbol{\theta}_{\text{MAP}}$ fixed. We denote this model as DPG-MIMO-E. All MIMO models use $C = 4$: $C = 8$ offered negligible performance gains and posed (partially stochastic) inference challenges, while $C = 2$ was less performant than $C = 4$.

**Metrics.** To provide summaries of our models' predictive accuracy and quality of uncertainty we use two proper scoring rules: the Negative Log-Likelihood (NLL) $p(\tilde{\boldsymbol{a}} \mid \tilde{\boldsymbol{e}}, \mathcal{T})$ and the Brier Score $\frac{1}{|\mathcal{E}|}\mathbb{E}_{\boldsymbol{\Theta}|\mathcal{T}}[\|\hat{\boldsymbol{p}}(\tilde{\boldsymbol{a}} \mid \tilde{\boldsymbol{e}}, \boldsymbol{\Theta}) - \tilde{\boldsymbol{a}}\|^2]$. Beyond proper scoring rules, we use Expected Calibration Error (ECE) (Guo et al., 2017), which measures the correspondence between predicted probabilities and empirical accuracy, and Error, defined as the percentage disagreement between a thresholded pred. mean $\mathbb{E}_{\boldsymbol{\Theta}|\mathcal{T}}[\tilde{\boldsymbol{a}} \mid \tilde{\boldsymbol{e}}, \mathcal{T}] > 0.5$ and the actual label $\tilde{\boldsymbol{a}}$. See Appendix A.2 for full details.

**Hyperparameters and inference details.** Utilizing NumPyro's NUTS implementation of HMC (Phan et al., 2019), our experiments run 4 chains in parallel, each chain taking 500 warm-up steps before generating 1000 samples, accumulating $M = 4000$ total samples. We use depth $D = 200$ for all models unless otherwise specified, as we did not observe significant improvements in predictive performance beyond this depth. We choose $\boldsymbol{a}_0 = \frac{1}{2} \cdot \mathbf{1}$ (reflecting prior uncertainty of existent edges) and $\boldsymbol{\lambda}_0 = 17 \cdot \mathbf{1}$ (approximate average value of limiting $\boldsymbol{\lambda}$ when running Algorithm 1 to convergence on $\text{RG}_{\frac{1}{3}}$ analytical distance matrices for $\theta = 1$) for all experiments. We did not find performance to be sensitive to such choices. Further details can be found in Appendix A.2.

## 6.2 Synthetic data evaluation

**Generative smooth signal model.** We build on (Dong et al., 2016)'s work on probabilistic smooth graph signal generation to validate our methods using synthetic data. Let $\boldsymbol{L} = \boldsymbol{U}\boldsymbol{\Lambda}\boldsymbol{U}^\top$ be the eigendecomposition of the graph Laplacian $\boldsymbol{L} = \text{diag}(\boldsymbol{A}\mathbf{1}) - \boldsymbol{A}$, with diagonal eigenvalue matrix $\boldsymbol{\Lambda}$ having associated eigenvalues $\lambda_i$ sorted in increasing order, $\boldsymbol{u}_i$ ($i$-th column of $\boldsymbol{U}$) is the eigenvector of $\boldsymbol{L}$ with eigenvalue $\lambda_i$. Further take $\boldsymbol{L}^\dagger$ to be the Moore-Penrose pseudoinverse of $\boldsymbol{L}$, with eigenvalues $\lambda_i^\dagger := \lambda_i^{-1}$ when $\lambda_i > 0$, and $\lambda_i^\dagger := 0$ otherwise. (Dong et al., 2016) proposed that smooth signals can be generated from a colored Gaussian distribution as $\boldsymbol{x} = \boldsymbol{\mu} + \sum_i \hat{x}_i \boldsymbol{u}_i$, where $\hat{x}_i \sim \mathcal{N}(0, \lambda_i^\dagger)$ and $\boldsymbol{\mu} \in \mathbb{R}^N$ denotes an arbitrary mean vector. Therefore $\boldsymbol{x} \sim \mathcal{N}(\boldsymbol{\mu}, \boldsymbol{L}^\dagger)$. To sample such a distribution, it suffices to draw an initial non-smooth signal $\boldsymbol{x}_0 \sim \mathcal{N}(\mathbf{0}, \boldsymbol{I})$

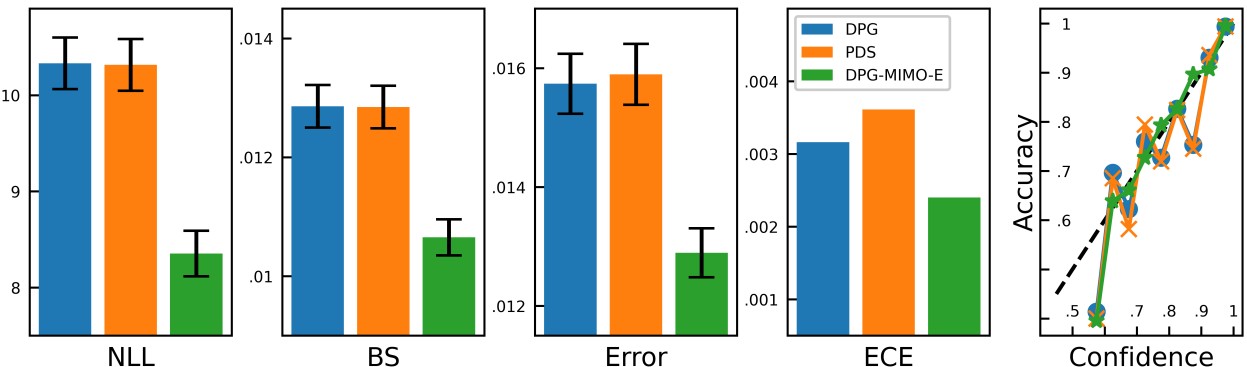

Figure 4: Effective i.i.d. generalization. Both DPG and PDS are performant and well calibrated BNNs, although PDS requires $\approx 3\times$ more time for inference. Further performance gains are found with the expanded, partially stochastic DPG-MIMO-E model. The rightmost plot (reliability diagram) indicates high calibration across confidence levels; the left-most bin is least calibrated but contains $< 0.4\%$ of edges across all models. This plot shows experiments on $N = 20$ RG$_{\frac{1}{3}}$ graphs. Error bars are scaled by .05 for compact visual effect.

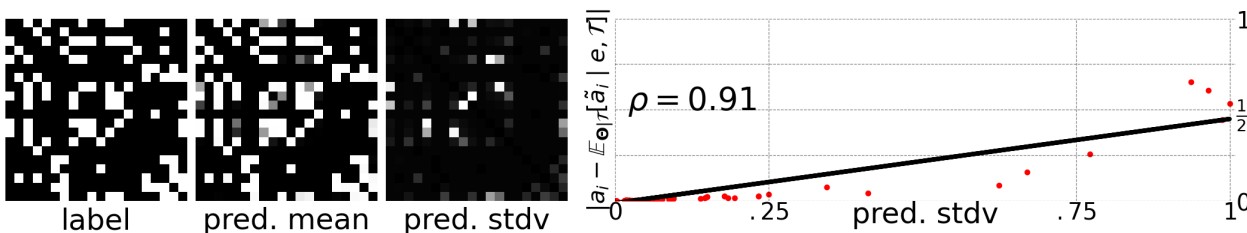

Figure 5: Qualitative i.i.d. generalization. *Left:* For a random test sample we show the label $\tilde{\boldsymbol{a}}$, and estimated mean (pred. mean) and standard deviation (pred. stdv) of the edge-wise marginal posterior predictive $p(\tilde{a}_i \mid \tilde{\boldsymbol{e}}, \mathcal{T})$. Comparing pred. mean to the label $\tilde{\boldsymbol{a}}$ adds qualitative evidence that the model is well fit to the data. *Right:* The edge-wise uncertainty estimate (pred. stdv.) and error $|\tilde{a}_i - \mathbb{E}_{\boldsymbol{\Theta}|\mathcal{T}}[\tilde{a}_i|\tilde{\boldsymbol{e}}, \mathcal{T}]|$ have a strong positive Pearson correlation $\rho = 0.91$.

and then compute $\boldsymbol{x} \leftarrow \boldsymbol{\mu} + \sqrt{\boldsymbol{L}^\dagger}\boldsymbol{x}_0$. When $\boldsymbol{\mu} = \boldsymbol{0}$, this procedure produces signals such that the power on the $i$-th frequency component is $\hat{x}_i^2 \sim \Gamma(k = \frac{1}{2}, \theta = 2\lambda_i^\dagger)$. Thus the expected power $\mathbb{E}[\hat{x}_i^2] = \lambda_i^\dagger$ is inversely proportional to the associated eigenvalue, concentrating energy on the low frequencies of the Laplacian spectrum. We construct the data matrix $\boldsymbol{X} = [\boldsymbol{x}_1, \ldots, \boldsymbol{x}_P] \in \mathbb{R}^{N \times P}$, where $\boldsymbol{x}_p$ are drawn i.i.d. from $\mathcal{N}(\boldsymbol{0}, \boldsymbol{L}^\dagger)$ as described above. Recalling the definition of the Euclidean distance matrix $\boldsymbol{E}$ in Section 3 and using the fact that $\bar{\boldsymbol{x}}_i^\top \bar{\boldsymbol{x}}_i \sim \Gamma(k = \frac{P}{2}, \theta = 2L_{ii}^\dagger)$ and $\mathbb{E}[\bar{\boldsymbol{x}}_i^\top \bar{\boldsymbol{x}}_j] = \sum_{p=1}^P \mathbb{E}[x_{ip}x_{jp}] = \sum_{p=1}^P \text{Cov}(x_{ip}, x_{jp}) = L_{ij}^\dagger, \forall(i, j)$, we can show $\mathbb{E}[\boldsymbol{E}] = \boldsymbol{1}\text{diag}(\boldsymbol{L}^\dagger)^\top + \text{diag}(\boldsymbol{L}^\dagger)\boldsymbol{1}^\top - 2\boldsymbol{L}^\dagger$ which we call the 'analytic distance matrix'. We henceforth use $\mathbb{E}[\boldsymbol{E}]$ and its finite sample approximations from $P$ signals for evaluation.

**Evaluation of i.i.d. generalization.** Here, we train and test DPG, PDS, and DPG-MIMO-E on $N = 20$ RG$_{\frac{1}{3}}$ graphs and their corresponding analytic distance matrices. We use $T = 50$ graphs for training and $100$ for testing. We present the results in Figure 4. DPG and PDS perform comparably; although DPG has marginally better accuracy and calibration. Significant improvement is found in the more expressive DPG-MIMO-E across all metrics, showing our simple 3-parameter model can be effectively expanded. These models are similarly performant across graph distributions (see the experiments reported in Table 1 and also in Appendix A.2); we show a single graph ensemble here for brevity.

Figure 5 depicts estimates of the first two moments of the posterior predictive distributions produced by DPG on a random test sample. We observe well-calibrated confidence and uncertainty, plus a strong correlation between error magnitude and uncertainty; indeed a useful uncertainty estimate should be a proxy for error.

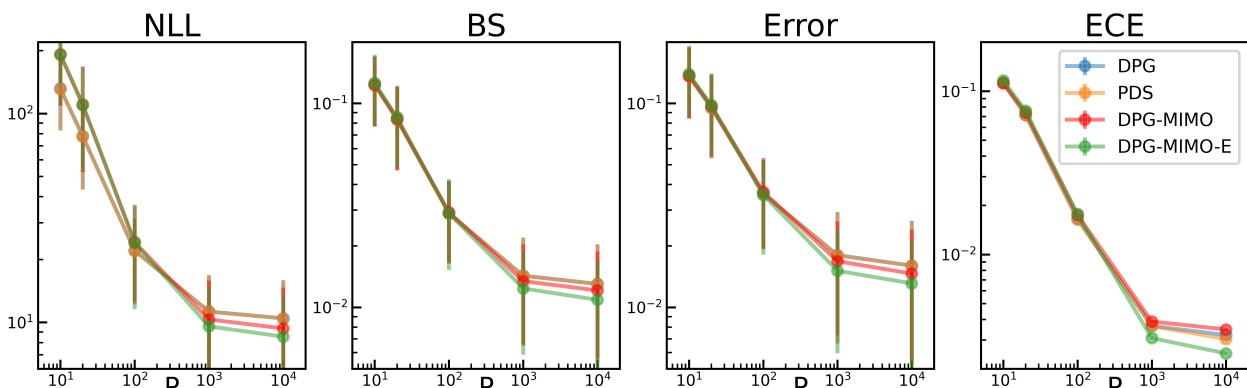

Figure 6: Detection of covariate shift across models. Using $\mathrm{RG}_{\frac{1}{3}}$ graphs with $N = 20$ nodes, we fit models on analytic Euclidean distance matrices and evaluate on corrupted distance matrices in a test set. Fewer signals $P$ used tends to increase corruption magnitude. Log-scaling unintentionally makes error bars appear to grow with $P$.

**Predictive uncertainty under distribution shift.** We evaluate two forms of distribution shift: corruptions to the noiseless analytic distance matrix $\mathbb{E}[\boldsymbol{E}]$ and label mismatch. To investigate corruptions, we train using $T = 50$ $\mathrm{RG}_{\frac{1}{3}}$ ($N = 20$) labels and their analytic distance matrix, and test on 100 $\mathrm{RG}_{\frac{1}{3}}$ samples which instead use $P$ signals to compute an Euclidean distance matrix $\boldsymbol{E}$. Fewer samples $P$ tend to produce larger corruption magnitudes, i.e., deviations from $\mathbb{E}[\boldsymbol{E}]$. Figure 6 shows that indeed, all models display lower predictive accuracy and higher uncertainty on the increasingly shifted input data, with the simpler models outperforming in the presence of larger corruption. To investigate label mismatch, models are fit to the same training data as above - $T = 50$ $\mathrm{RG}_{\frac{1}{3}}$ ($N = 20$) labels with analytic distance matrix input - but now tested on 100 test samples from the following *different* $N = 20$ random graph distributions: (i) random geometric graphs with connectivity radius $\frac{1}{2}$ ($\mathrm{RG}_{\frac{1}{2}}$), Erdős-Rényi graphs with edge probability $p = 0.5$ ($\mathrm{ER}_{\frac{1}{2}}$), and Barabási-Albert graphs with $m = 1$ link per node ($\mathrm{BA}_1$). Results are displayed in Table 1. All models show lower accuracy ($\uparrow$ Brier Score) and commensurately higher uncertainty ($\uparrow$ NLL) on these mismatched data. DPG methods tend to maintain better calibration than PDS.

Table 1: Detection of label mismatch. Models are fit to $\mathrm{RG}_{\frac{1}{3}}$ and tested on 100 samples from the same $\mathrm{RG}_{\frac{1}{3}}$ as well as three *different* graph distributions. For each metric, the mean over the test set is reported.

| | NLL | | | | BS ($\times 10^{-2}$) | | | | ECE ($\times 10^{-3}$) | | | |
|---|---|---|---|---|---|---|---|---|---|---|---|---|
| | $\mathrm{RG}_{\frac{1}{3}}$ | $\mathrm{RG}_{\frac{1}{2}}$ | $\mathrm{ER}_{\frac{1}{2}}$ | $\mathrm{BA}_1$ | $\mathrm{RG}_{\frac{1}{3}}$ | $\mathrm{RG}_{\frac{1}{2}}$ | $\mathrm{ER}_{\frac{1}{2}}$ | $\mathrm{BA}_1$ | $\mathrm{RG}_{\frac{1}{3}}$ | $\mathrm{RG}_{\frac{1}{2}}$ | $\mathrm{ER}_{\frac{1}{2}}$ | $\mathrm{BA}_1$ |
| DPG | 10.33 | 17.53 | 33.01 | 20.91 | 1.29 | 2.29 | 4.66 | 2.10 | 3.42 | 52.06 | 171.38 | 16.60 |
| PDS | 10.32 | 17.70 | 33.43 | 20.80 | 1.29 | 2.33 | 4.80 | 2.10 | 3.75 | 52.87 | 173.53 | 16.60 |
| DPG-MIMO | 9.19 | 17.13 | 40.49 | 13.37 | 1.19 | 1.74 | 4.62 | 2.06 | 3.50 | 54.61 | 161.71 | 11.24 |
| DPG-MIMO-E | 8.35 | 12.66 | 37.77 | 8.35 | 1.07 | 1.52 | 7.82 | 1.32 | 2.40 | 29.17 | 29.07 | 2.84 |

**Scaling to larger graphs.** Direct inference in large graph settings is challenged by substantial memory demands. Specifically, providing gradients for HMC via naive backpropagation necessitates storing all intermediate outputs, while full Bayesian inference mandates holding the full data set in memory, cumulatively leading to a memory footprint of $\mathcal{O}(DN^2T)$. Adopting forward-mode auto-differentiation partially mitigates this issue by removing depth dependency, effectively reducing memory complexity to $\mathcal{O}(N^2T)$. The time complexity for MCMC also presents scalability challenges: each forward pass has time complexity of $\mathcal{O}(DN^2T)$, and each posterior sample requires multiple forward passes. Attempting to instead perform transfer learning – where inference occurs with smaller graphs to then test on larger graphs – still faces an obstacle. The objective function (3) contains the terms $2\boldsymbol{a}^\top \boldsymbol{e} = 2\sum_{i=1}^{N(N-1)/2} a_i e_i$, $\frac{\beta}{2}\|\boldsymbol{a}\|_2^2 = \frac{\beta}{2}\sum_{i=1}^{N(N-1)/2} a_i^2$, and $-\alpha \mathbf{1}^\top \log(\boldsymbol{Sa}) = -\alpha \sum_{i=1}^{N} \log(\boldsymbol{Sa})_i$. The former two terms involve $N(N-1)/2$ summands, while the

latter is a sum over $N$ elements. This results in disparate growth rates for these terms as $N$ increases, which poses a challenge for parameter optimization across different graph sizes, i.e., we find parameters optimized for a graph with $N_i$ nodes are not effective for $N >> N_i$. Moreover, the differing growth rates cause standard cardinality-based normalization schemes (dividing each term by its number summands) to fail, and determining analytically how the parameters should change as a function of size is non-trivial; see Appendix A.3 for a detailed discussion.

In light of this, we perform an adjusted transfer learning experiment by fitting the empirical growth trends of MAP DPG parameter estimates on $N = \{20, 50, 100, 200\}$ $\mathrm{ER}_{\frac{1}{4}}$ graphs and their corresponding analytic distance matrices. By using a moderate depth $D = 200$ and $T = 10$ training samples, MAP inference on larger graphs with $N = 200$ nodes takes $\sim 1$ hour on a M2 MacBook laptop. Using this empirical parameter fit we can extrapolate parameter values and perform transfer learning on 100 $\mathrm{ER}_{\frac{1}{4}}$ test graphs of size up to $N = 1000$; again done locally without any GPU. The transfer learning experiment reveals an expected but graceful decay in performance in NLL as $N$ increases. Further information and plots on these scaling experiments are available in Appendix A.3.

### 6.3 Ablation studies

**Prior modeling.** To inspect the influence of informative prior modeling on DPG, we define model 'DPG-U' which replaces DPG's informative priors with the following uninformative priors: $\log \theta \sim U\left[10^{-6}, 10^6\right]$, $\delta, b \sim \mathcal{N}(\mathbf{0}, 10^3 \mathbf{I})$. What we find is that in data-rich regimes an informative prior mostly acts to improve efficiency of posterior inference, e.g., when fitting to $T = 50$ $\mathrm{RG}_{\frac{1}{3}}$ graphs with $N = 20$ nodes (using analytic distance matrices) and testing on 100 i.i.d. samples, it reduces the time needed to generate $M = 4000$ samples by a factor of $7.8\times$ and increases the effective sample size - a measure of the number of independent samples with the same estimation power as the observed correlated samples (Gelman et al., 1995) - for $\theta$ ($\mathrm{ESS}_\theta$) by a factor of $5\times$. In data poor regimes it also tends to help slow down performance degradation, e.g., when instead using $T = 2$ with the same graph data, NLL and ECE are 37% and 13.7% better with the prior than without. See Table 3 in Appendix A.4 for the full numerical results.

**Partial stochasticity.** We conducted an ablation study investigating the benefits of adding partial stochasticity to the DPG-MIMO model ($C = 4, D = 200$) using MAP point estimates; the partially stochastic setup is described in Section 6.1. The training and test sets are identical to the i.i.d. generalization experiment above. Our experiments showed that introducing partial stochasticity in DPG-MIMO-E markedly improved NLL, BS, error, and ECE by 9.1%, 10.3%, 9.0%, and 31.4%, respectively.

### 6.4 Real data evaluation

**Quantifying uncertainty in networks learnt from S&P500 stock prices.** To verify that DPG provides useful measures of uncertainty on edge predictions based on real data, we first use the financial dataset presented in (de Miranda Cardoso et al., 2021). The time series consist of S&P500 daily stock prices from three sectors (Communication Services, Utilities, and Real Estate), comprising a total of $N = 82$ stocks. The data spans the period from Jan. 3rd 2014 to Dec. 29th 2017, yielding $P = 1006$ daily observations. We divided these stocks equally into training and testing sets. For both sets, we created a log-returns data matrix, denoted as $\boldsymbol{X} \in \mathbb{R}^{N/2 \times P}$. Specifically, $X_{i,j} = \log SP_{i,j} - \log SP_{i,j-1}$, where $SP_{i,j}$ signifies the closing price of the $i$-th stock on the $j$-th day. From each matrix (train and test), we derived a matrix of Pearson correlation coefficients $\boldsymbol{\Sigma}$. We take the input to be $\boldsymbol{E} \leftarrow \mathbf{1}\mathbf{1}^\top - |\boldsymbol{\Sigma}|$, thus yielding measure of dissimilarity. We also constructed a binary label graph, assigning an edge weight value of 1 for stocks within the same sector and 0 for pairs in different sectors. Thus, our training and testing datasets each contain a single sample. Nodes from the same sector are numbered contiguously creating a block diagonal adjacency matrix structure, displayed with the red outline in Figure 7 (left).

We use DPG with depth $D = 200$ and identify a high-performing parameter value range for $\theta$ via predictive checking as in Section 5.2. Such values aligned closely with those found in the synthetic experiments, and so we keep the priors unchanged. We run Bayesian parameter inference on a M2 MacBook laptop which takes about 2 minutes; we visualize the input, empirical mean, and standard deviation of the posterior predictive on the test sample in Figure 7 (left). Notably, while our mean recovery is robust, there are areas where it

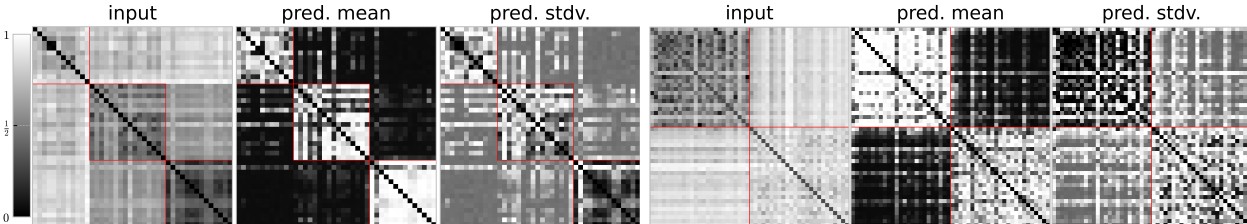

Figure 7: Quantifying Uncertainty with Financial Data and Images of Digits. *Left*: S&P500. We randomly split stocks from 3 sectors of the S&P500 and compute their input $\mathbf{1}\mathbf{1}^\top - |\mathbf{\Sigma}|$, where $\mathbf{\Sigma}$ is the matrix of Pearson correlation coefficients, using log daily returns over an extended period. The graph label connects stocks of the same sector indicated by red block diagonal outline. There is only a single train and test sample. Here, we display the test sample input and estimates of the mean (pred. mean) and standard deviation (pred. stdv.) of the posterior predictive. In pred. mean, white (black) inside (outside) the block diagonal indicates correct prediction for both experiments. Pred. mean which don't match their label tend to have larger uncertainty. Error and pred. stdv. have a Pearson correlation of 0.70 over the test set. *Right*: MNIST digits. We construct graphs of MNIST digits "1" and "2", connecting nodes of the same digit, similarly outlined in red ("1"s are the first block). The input is the log pairwise Euclidean distance of their vectorized image. Notice "1"s tend to be much closer to each other than "2"s. DPG again shows an ability to place higher uncertainty on edges with higher error. Error and pred. stdv. have a Pearson correlation of 0.62 over the test set. Pred. stdv. in both plots are maximum normalized for visual clarity.

deviates from the label. These areas tend to exhibit increased variation in the posterior predictive. The Pearson correlation coefficient between the error and predictive standard deviation on all edges, true positive edges, and true negative edges is 0.70, 0.70, and 0.79, respectively, suggests that our model's uncertainty can be a useful proxy for error in the absence of labels in this financial test case.

**Learning the graph of MNIST digits 1 and 2.** To further demonstrate DPG's capability to quantify uncertainty on graphs estimated from images of digits, we emulate the experiment from (Kalofolias, 2016) which learns a graph to classify handwritten digits "1" and "2" from the MNIST database (LeCun et al., 1998). Each digit is an image of $28 \times 28$ pixels, each pixel taking integer value in $0, \ldots, 255$. As (Kalofolias, 2016) reports, this problem is particular because digits "1" are much close to each other than "2" are (average square distance of 45 and 102, respectively). A single sample in our dataset is constructed as follows. We randomly sample 25 images of digits "1" and "2". Each image represents a node in this graph sample, hence $N = 50$. We connect nodes of the same digit with a binary edge. The image corresponding to each node is vectorized into $\bar{\boldsymbol{x}}_i \in \mathbb{R}^{28^2}$, $i = 1, \ldots N$, becoming the nodal feature vector. For ease of label visualization, we order the nodes such that all "1" digits come before "2" digits, which creates a block diagonal adjacency matrix, indicated by the red outline in Figure 7 (right). The nodal dissimilarity matrix $\boldsymbol{E}$ is computed as the log pairwise Euclidean distance of the node features (vectorized digit images). We construct a training and test set of size $T = 5$ and 50, respectively.

We use the same modeling and inference setup as in the previous real data experiment; inference takes $\approx 10$ minutes. Figure 7 (right) displays the input $\boldsymbol{E}$, pred. mean, and pred. stdv. on a random test sample. Visual inspection reveals performant mean prediction, and that edges with high errors tend to have high variation. Indeed, across the entire test set the two have a strong positive Pearson correlation coefficient over all edges, true positive edges, and true negative edges of 0.62, 0.72, and 0.51, respectively, suggesting DPG effectively quantifies predictive uncertainty of edges in this image analysis setting.

# 7 Concluding summary, limitations, and the road ahead

In this work we identify independent interpretability of (regularization) parameters in inverse problems as a key property, which allows one to incorporate prior information on solution characteristics into prior distributions over the NN parameters of an unrolled optimization algorithm. We investigate an inverse problem with this independent interpretability property in the context of GSL from smooth signals, and

introduce an optimization algorithm that is simpler than existing methods, without sacrificing estimation performance. We unroll this algorithm producing the first true NN for GSL from smooth signals. We leverage independent interpretability to incorporate prior information about sparsity characteristics of sought graphs into prior distributions over unrolled network parameters, producing the first BNN for GSL from smooth signal observations. We lean into the advantages of unrollings - low parameter dimensionality, fast compiler and auto-grad friendly layers, as well as the empirical need for shallow networks - to perform posterior approximation using MCMC sampling. Doing so yields high-quality and well-calibrated uncertainty estimates over edge predictions, as demonstrated via comprehensive experiments with synthetic and real datasets.

The primary limitation of our approach lies in its scalability to moderate- and large-sized graphs and datasets. The computational and memory requirements to achieve asymptotically exact posterior inference through MCMC sampling are substantial, owing to the necessity for numerous forward network passes per posterior sample and the requirement to store the entire data set in memory. Promising directions of future work include exploration of non-asymptotically exact posterior inference methods, namely variational inference, which typically requires much less computation and allows for mini-batch training and thus use of larger datasets (Jospin et al., 2022). As we produce distributions over output graphs, future work can explore Bayesian decision analysis with the introduction of a utility function, or even using DPG as a module within a larger system, where propagation of uncertainty over the inferred graph is important.

### Acknowledgments

Work in this paper was supported in part by the NSF under award ECCS-2231036. The authors would like to thank Canberk Ekmekci for fruitful discussions during the early stages of this project.

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

# A   Appendix

## A.1   Optimization algorithms for model-based GSL methods

### A.1.1   Primal-dual splitting (PDS) algorithm

Algorithm 2 was introduced in (Kalofolias, 2016) to solve the $(\alpha, \beta)$-parameterization of (3), and was subsequently unrolled by (Pu et al., 2021) on the same objective. Note the increased number of operations, intermediate variables, and parameters as compared to Algorithm 1.

---
**Algorithm 2** Proximal Dual Splitting (PDS)

---
**Inputs**: Fixed parameters $\alpha, \beta, \gamma \in \mathbb{R}$, and data $\boldsymbol{e}$.
**Initialize**: $\boldsymbol{a}_0$ and $\boldsymbol{v}_0$ at random.
**for** $k = 1, 2, \ldots$ **do**
$\quad \boldsymbol{r}_{1,k} = \boldsymbol{a}_k - \gamma(2\beta\boldsymbol{a}_k + 2\boldsymbol{e} + \boldsymbol{S}^\top\boldsymbol{v}_k).$
$\quad \boldsymbol{r}_{2,k} = \boldsymbol{v}_k + \gamma\boldsymbol{S}\boldsymbol{a}_k.$
$\quad \boldsymbol{p}_{1,k} = \mathrm{prox}_{\gamma,\Omega_1}(\boldsymbol{r}_{1,k}), \text{ where } \mathrm{prox}_{\gamma,\Omega_1}(\boldsymbol{r}_{1,k}) = \max\{0, \boldsymbol{r}_{1,k}\}.$
$\quad \boldsymbol{p}_{2,k} = \mathrm{prox}_{\gamma,\Omega_2}(\boldsymbol{r}_{2,k}), \text{ where } \left(\mathrm{prox}_{\gamma,\Omega_2}(\boldsymbol{r}_{2,k})\right)_i = (r_{2_i} - \sqrt{r_{2_i}^2 + 4\alpha\gamma})/2.$
$\quad \boldsymbol{q}_{1,k} = \boldsymbol{p}_{1,k} - \gamma(2\beta\boldsymbol{p}_{1,k} + 2\boldsymbol{e} + \boldsymbol{S}^\top\boldsymbol{p}_{2,k}).$
$\quad \boldsymbol{q}_{2,k} = \boldsymbol{p}_{2,k} - \gamma\boldsymbol{S}\boldsymbol{p}_{1,k}.$
$\quad \boldsymbol{a}_{k+1} = \boldsymbol{a}_k - \boldsymbol{r}_{1,k} + \boldsymbol{q}_{1,k}.$
$\quad \boldsymbol{v}_{k+1} = \boldsymbol{v}_k - \boldsymbol{r}_{2,k} + \boldsymbol{q}_{2,k}.$
**end for**
**Return**: $\boldsymbol{a}_{k+1}$

---

**Parameter tuning in the convergent setting.** The optimization parameters $\alpha, \beta$ and $\gamma$ of Algorithm 2 are not interpretable, and so performing parameter tuning incurs a $\mathcal{O}(K^3)$, where $K$ is the number of discretization values used for each parameter. This is in contrast to Algorithm 1 which has only two parameters $\theta$ and $\delta$. Since $\theta$ is independently interpretable w.r.t sparsity, we can first tune $\theta$ to produce outputs with desired sparsity level, then tune $\delta$ for appropriate edge weight scale, reducing reducing tuning costs to $\mathcal{O}(K)$.

**Step size $\gamma$ frustrates Bayesian PDS.** Algorithm 2 has a nuisance step size parameter $\gamma$ which must be properly tuned for convergent and performant iterations; problematically such $\gamma$ values are a functions of $\alpha$ and $\beta$ values; see (Kalofolias, 2016). The step size $\gamma$ introduces several issues for use as a learned parameter in a BNN. First, we empirically find that the $\gamma$ values which produce convergent iterations of Algorithm 2 (within $5 \times 10^4$ iterations) are very close to $\gamma$ values which produce divergent iterations, as shown in Figure 8 (left). This presents the practical problem of producing NaNs during Bayesian inference, as the sampler will be drawn toward such unstable values of $\gamma$, causing the unrolling to diverge. Second, the values of $\gamma$ which produce divergent unrollings are themselves a function of $\alpha$, $\beta$, and the unrolling depth, frustrating simple solutions to prevent divergence, e.g., setting $p(\gamma)$ to be some fixed closed interval. Third, the products $\gamma\beta$ and $\gamma\alpha$ ensure the Unrolled PDS is *not a neural network*; this causes problems in practice as products of parameters - each of which can vary over many orders of magnitude - can cause problematic gradients. These observations help explain why we could not find any prior configuration over the three parameters $\{\alpha, \beta, \gamma\}$ in PDS which produced convergent posterior sampling over multiple depths. Fixing $\gamma$ to a scalar value ameliorates these three issues, indeed this was the only configuration which produced a performant

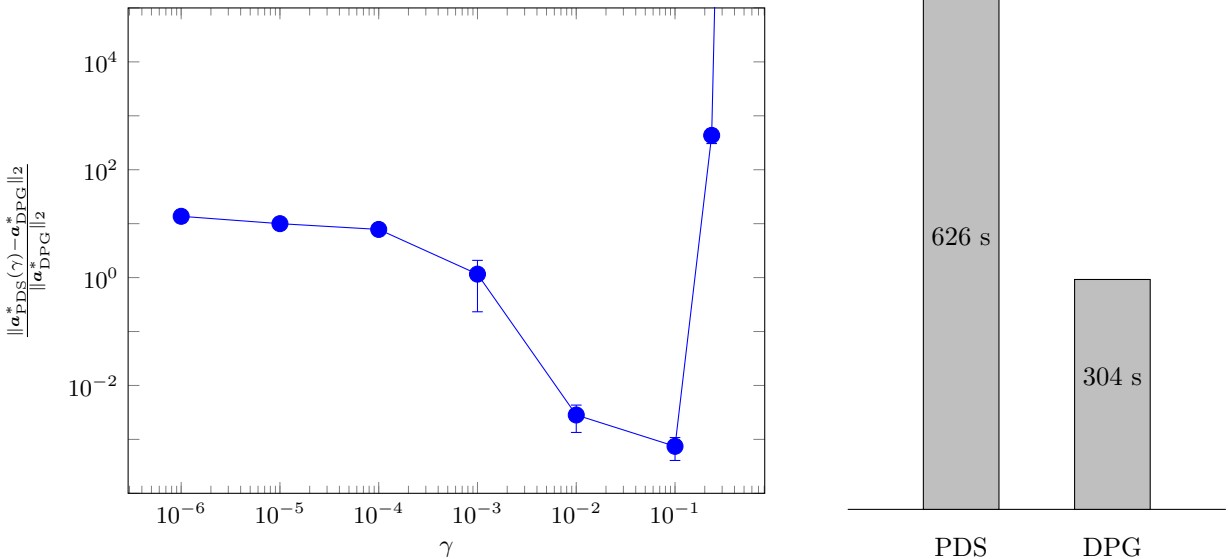

Figure 8: *Left*: PDS Step Size $\gamma$. Running both DPG and PDS iterations with $\alpha = \beta = 1$ and $\theta = 1/\sqrt{\alpha\beta} = 1$, $\delta = \sqrt{\alpha/\beta} = 1$ for $5 \times 10^4$ iterations, we plot the mean and standard deviation of the normalized $\ell_2$ distance between the output of PDS vs DPG. The value of $\gamma$ resulting in the most faithful PDS solution is close to values which yield divergent iterations, problematic for its use as our NN function. *Right*: PDS vs DPG inference time using $D = 200$ on $T = 50$ $\mathrm{RG}_{\frac{1}{3}}$ graphs running NUTS with 4 chains (in parallel), each taking $M = 1500$ samples.

PDS model. We found $\gamma = 0.1$ worked best. Even so, care has to be taken in setting the priors for $\alpha$ and $\beta$: $\log \alpha$ and $\log \beta \sim U\left[10^{-6}, 10^6\right]$ produced all divergent paths - recall this naive setting for $\log \theta$ in DPG worked with no issue in the Ablation Studies of Section 6.3. A successful avenue we followed to produce a performant PDS model was to use $\alpha, \beta \sim \mathrm{LogNormal}(0, 10)$ and $b \sim \mathcal{N}(0, 10^3)$.

**Comparing Bayesian inference time of PDS vs DPG.** Above we show that finding a range of reasonable values of the optimisation parameters can be cubic in PDS while linear in DPG, owing to independent interpretability. Further, PDS takes more than twice as long in performing inference as the DPG (and PDS with stochastic $\gamma$ took an order of magnitude longer). This may be due to the increased complexity of PDS iterates and the fact that both $\alpha$ and $\beta$ occur inside PDS iterations, compared to only $\theta$ for DPG iterates, increasing the complexity of the computational graph required to compute parameter gradients. See Figure 8 (right) showing the difference in inference runtime for PDS and DPG, both having depth $D = 200$ using $T = 50$ $\mathrm{RG}_{\frac{1}{3}}$ graphs.

## A.2 Experimental Details

### A.2.1 Scoring rules and evaluation metrics

Fitting the model to training data $\mathcal{T}$ produces samples $\{\boldsymbol{\Theta}^{(m)}\}_{m=1}^M$ from the posterior $p(\boldsymbol{\Theta} \mid \mathcal{T})$. Given test sample $\{\tilde{\boldsymbol{e}}, \tilde{\boldsymbol{a}}\}$ we can evaluate the quality of fit using proper scoring rules, e.g., Log Predictive Density and Brier Score, and discrete metrics, e.g., the Error.

**Log predictive density, i.e., Log Likelihood.** The Log Likelihood measures the quality of the probabilistic predictions of the model. It is defined as the predicted probability of the true outcome under the model.

$$\log p(\tilde{\boldsymbol{a}} \mid \tilde{\boldsymbol{e}}, \mathcal{T}) = \log \int p(\tilde{\boldsymbol{a}} \mid \tilde{\boldsymbol{e}}, \boldsymbol{\Theta}) p(\boldsymbol{\Theta} \mid \mathcal{T}) d\boldsymbol{\Theta}$$

$$\approx \log \left( \frac{1}{M} \sum_{m=1}^{M} p(\tilde{\boldsymbol{a}} \mid \tilde{\boldsymbol{e}}, \boldsymbol{\Theta}^{(m)}) \right)$$

$$= \log \sum_{m=1}^{M} p(\tilde{\boldsymbol{a}} \mid \tilde{\boldsymbol{e}}, \boldsymbol{\Theta}^{(m)}) - \log S$$

$$= \text{log-sum-exp} \left\{ \log p(\tilde{\boldsymbol{a}} \mid \tilde{\boldsymbol{e}}, \boldsymbol{\Theta}^{(1)}), \dots, \log p(\tilde{\boldsymbol{a}} \mid \tilde{\boldsymbol{e}}, \boldsymbol{\Theta}^{(M)}) \right\} - \log S \qquad (9)$$

The NLL is simply $-1 \cdot \log p(\tilde{\boldsymbol{a}} \mid \tilde{\boldsymbol{e}}, \mathcal{T})$. A lower NLL indicates better predictive performance, as it suggests that the model assigns higher probabilities to the observed outcomes.

**Brier Score.** The Brier Score is used to assess the accuracy of probabilistic predictions. It measures the mean squared difference between the predicted probability and the actual outcome. A lower Brier Score indicates better calibration and accuracy of the probabilistic predictions. Recall the edge-wise probabilities output by the model for parameters $\boldsymbol{\Theta}^{(m)}$ are denoted $\hat{\boldsymbol{p}}^{(m)} = \sigma(\delta^{(m)} \Gamma_{\theta^{(m)}}^{D}(\boldsymbol{e}) - b^{(m)})$. Then the Brier Score is

$$\text{BS}(\tilde{\boldsymbol{a}} \mid \tilde{\boldsymbol{e}}, \mathcal{T}) = \frac{1}{|\mathcal{E}|} \int \|\hat{\boldsymbol{p}} - \tilde{\boldsymbol{a}}\|_2^2 \cdot p(\boldsymbol{\Theta} | \mathcal{T}) d\boldsymbol{\Theta}$$

$$\approx \frac{1}{M|\mathcal{E}|} \sum_{m=1}^{M} \|\hat{\boldsymbol{p}}^{(m)} - \tilde{\boldsymbol{a}}\|_2^2 \qquad (10)$$

*Complementary metrics.* The NLL is particularly useful for evaluating the overall fit of a probabilistic model, emphasizing the accuracy of the assigned probabilities, especially for less likely events. In contrast, the Brier Score provides a more intuitive measure of predictive accuracy and calibration, making it easier to interpret the model's probabilistic predictions in practical scenarios.

*Numerical Concerns.* Evaluating $\log p(\boldsymbol{a}|\boldsymbol{e}, \boldsymbol{\Theta}^{(m)})$ can cause numerical issues which stem from underflow in computing $\log \hat{a}^{(m)}$ or $\log(1 - \hat{a}^{(m)})$. for some edges. This becomes an issue in the covariate shift experiments in Section 6, where the data becomes very noisy and the model can confidently predict the wrong label. When this underflow occurs, the log evaluates to $-\infty$. To solve this we use the softplus parametrization of the log likelihood: $-\text{softplus}(-\bar{y} \cdot (\delta^{(m)} \Gamma_{\theta^{(m)}}^{D}(\boldsymbol{e}) - b^{(m)}))$, where $\bar{y} = 2y - 1$, see Equation (10.13) in (Murphy, 2023).

**Error.** We take the error to be the percentage of incorrectly predicted edges between our thresholded pred. mean $\mathbb{E}_{\boldsymbol{\Theta}|\mathcal{T}}[\tilde{\boldsymbol{a}} \mid \tilde{\boldsymbol{e}}, \mathcal{T}] > 0.5$ and the true label $\tilde{\boldsymbol{a}}$.

**Calibration for GSL.** We follow the calibration procedure laid out in (Guo et al., 2017). To fit into this framework, we take our prediction to be the pred. mean thresholded at 0.5, i.e., $\mathbb{E}_{\boldsymbol{\Theta}|\mathcal{T}}[\tilde{\boldsymbol{a}} \mid \tilde{\boldsymbol{e}}, \mathcal{T}] > 0.5$. Thus the confidence, i.e., the probabilities associated with the predicted label, will thus always be in $\in [0.5, 1]$. We will thus construct $M$ uniform bins $I_m = (.5 + \frac{.5*(m-1)}{M}, .5 + \frac{.5*m}{M}]$, $m = \{1, \dots, M\}$. If an edge confidence is in $I_m$, we assign it to bin $B_m$. We then evaluate the accuracy $\text{acc}(B_m) := \frac{1}{|B_m|} \sum_{i \in B_m} \mathbf{1}(\hat{a}_i = a_i)$ and average confidence $\text{conf}(B_m) := \frac{1}{|B_m|} \sum_{i \in B_m} \hat{a}_i$ for each bin. The Expected Calibration Error (ECE) is defined as the weighted average the bins' accuracy/confidence difference:

$$\text{ECE} = \sum_{m=1}^{M} \frac{|B_m|}{B} |\text{acc}(B_m) - \text{conf}(B_m)|,$$

where $B := \sum_{m=1}^{M} |B_m|$ are the total number of edges predicted. Thus the ECE provides a measure of calibration over all edges in the evaluation set.

### A.2.2 Inference Details

**Details on HMC.** To ensure convergence has been achieved in our posterior sampling we simulate multiple

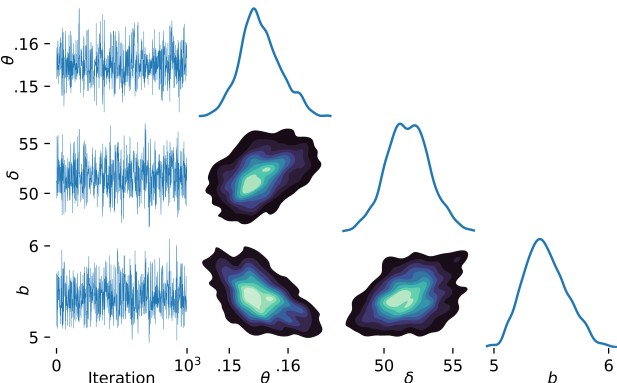

Figure 9: Trace plots and bivariate kernel density estimates of DPGs parameters on $\text{RG}_{\frac{1}{3}}$ graphs.

chains with different starting points in the parameter space, discard the first 500 iterations of each simulation, and monitor relevant diagnostic criteria - namely the Potential Scale Reduction Factor (PSRF) $\hat{r}$ and Effective Sample Size (ESS). We do not perform thinning on the resulting samples. Unless otherwise stated all inference was achieved without issue, namely with that $\hat{r} \approx 1$ and $\hat{n}_{eff} \geq 1000$ for each parameter. For a visualization of successful HMC inference with DPG, see Figure 9 which displays the trace plots and bivariate kernel density estimates of DPG with $D = 30$ on $\text{RG}_{\frac{1}{3}}$ graphs using analytic distance matrices.

**MAP vs MLE.** Due to nonconvexity in the loss, initialization is important in finding MAP and MLE of parameters. We did not alter the default initialization provided in NumPyro with MAP, as it worked without issue. We found MLE to be more dependant on initialization; we initialized the parameters of MLE models to the median value of the priors set in Section 4, i.e. $\theta, \delta, b \approx 0.32, 100, 1$.

**Log-normal parameterization.** Let $Z$ be a standard normal variable and let $\mu$ and $\sigma > 0$ be two real numbers, then the distribution of the random variable $X = e^{\mu + \sigma Z}$ is called a log-normal distribution with parameters $\mu$ and $\sigma$, i.e. $X \sim \text{Lognormal}(\mu, \sigma^2)$. Parameters $\mu$ and $\sigma$ represent the mean and standard deviation of $\log X$, not $X$ itself. While the normality of $\log X$ is true regardless of base, we will be performing modeling using powers of 10. Thus in the main body when we specify a distribution $\text{Lognormal}(\mu, \sigma^2)$, it implies the location parameter used is $\ln 10^{\mu}$, while the scale parameter is unchanged. For example, $\text{Lognormal}(2, 4)$ would imply location $\mu = \ln 10^2$ and scale $\sigma = 2$, and thus would be invoked as a call to the probabilistic programming language distribution, here NumPyro, as numpyro.distributions.LogNormal(loc $= \ln 10^2$, scale $= 2$).

### A.3 Scaling to Large Graphs

**Scaling via transfer learning in the convergent setting.** In (3), terms $2\boldsymbol{a}^{\top}\boldsymbol{e}, \frac{\beta}{2}\|\boldsymbol{a}\|_2^2$ have $N(N-1)/2$ components while term $-\alpha\mathbf{1}^{\top}\log(\boldsymbol{S}\boldsymbol{a})$ has $N$ components, thus the former terms will tend to dominate the latter as $N$ grows (using fixed $\alpha$, $\beta$). We thus create a new problem where each term is normalized by it's number of components; we use the notation $\bar{\alpha}, \bar{\beta}$ to indicate the *scale invariant* parameter values

$$\arg\min_{\boldsymbol{a}} \left\{ |\mathcal{E}_N|^{-1} 2\boldsymbol{a}^{\top}\boldsymbol{e} - N^{-1}\bar{\alpha}\mathbf{1}^{\top}\log(\boldsymbol{S}\boldsymbol{a}) + |\mathcal{E}_N|^{-1}\frac{\bar{\beta}}{2}\|\boldsymbol{a}\|_2^2 + \mathbb{I}\{\boldsymbol{a} \geq 0\} \right\} \tag{11}$$

$$= \boldsymbol{a}^*(|\mathcal{E}_N|^{-1}\boldsymbol{e}, N^{-1}\bar{\alpha}, |\mathcal{E}_N|^{-1}\bar{\beta}) \text{ , 1-1 correspondence between objective terms and params/inputs}$$

$$= \boldsymbol{a}^*(\boldsymbol{e}, |\mathcal{E}_N|N^{-1}\bar{\alpha}, \bar{\beta}) \qquad \text{, argmin invariant to positive scaling, multiply by } |\mathcal{E}_N|$$

$$= \boldsymbol{a}^*(\boldsymbol{e}, .5(N-1)\bar{\alpha}, \bar{\beta}). \tag{12}$$

Intuitively, (12) tells us to scale $\alpha$ by $\mathcal{O}(N)$. This makes sense, as $\alpha$ itself scales the objective term which grows at a factor of $\mathcal{O}(N)$ slower than the other (finite) terms. This should help correct the differing component sizes as the problem dimension $N$ grows. Eq. (12) formulates the solution to the size normalized problem with respect to the solutions of original problem (3). This shows that we can use the original solution procedures,

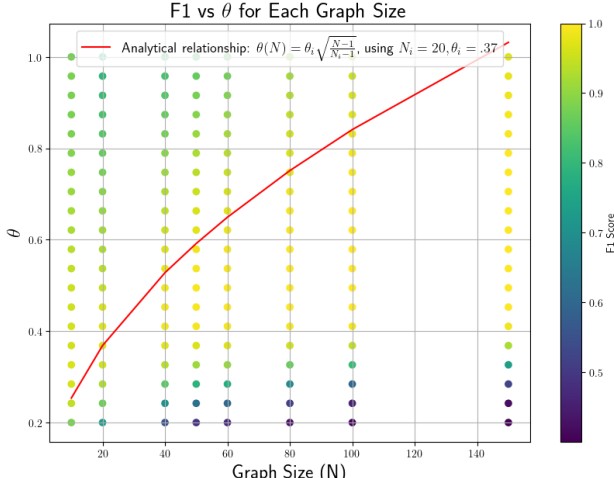

Figure 10: Transferring to Larger Graphs. Using $T = 20$ $ER_{\frac{1}{4}}$ graphs of varying sizes, we run a grid search to find optimal $\theta$ for the structure recovery task using Algorithm 1 for each graph size, and show the F1 score for each $\theta$. We also plot the line corresponding to Equation 13 which extrapolates performant $\theta$ values from the $N_i = 20$ setting. This line coincides with empirically observed highly performant $\theta$ values in problem setting for both smaller and significantly larger graphs.

now with scaled $\alpha$, to produce solutions. To do so, suppose we find optimal parameter values $\alpha_i$, $\beta_i$ to the original problem (3) with problem size $N_i$. To find the proper parameter values for a different problem size $N$, we can solve for the scale invariant parameter values, then re-scale to the appropriate size as

$$\alpha(N) = \overbrace{\alpha_i\big(.5(N_i - 1)\big)^{-1}}^{\bar{\alpha}}.5(N - 1)$$
$$\beta(N) = \underbrace{\beta_i}_{\bar{\beta}} .$$

Note as a sanity check, when $N = N_i$ we have $\alpha(N) = \alpha_i$ and $\beta(N) = \beta_i$. It is trivial to recover similar functions for $\theta(N)$ and $\delta(N)$ by substituting into the definitions of $\theta = \frac{1}{\sqrt{\alpha\beta}}$ and $\delta = \sqrt{\frac{\alpha}{\beta}}$:

$$\theta(N) = \frac{1}{\sqrt{\alpha(N)\beta(N)}} \quad = \overbrace{\theta_i\big(.5(N_i - 1)\big)^{\frac{1}{2}}}^{\bar{\theta}} \big(.5(N - 1)\big)^{-\frac{1}{2}} = \theta_i\sqrt{\frac{N_i - 1}{N - 1}} \tag{13}$$

$$\delta(N) = \sqrt{\alpha(N)\beta(N)^{-1}} = \underbrace{\delta_i\big(.5(N_i - 1)\big)^{-\frac{1}{2}}}_{\bar{\delta}}\big(.5(N - 1)\big)^{\frac{1}{2}} \quad = \delta_i\sqrt{\frac{N - 1}{N_i - 1}}. \tag{14}$$

See Figure 10 which empirically validates this derivation on $ER_{\frac{1}{4}}$ graphs across an order of magnitude difference in problem size $N$.

**Scaling in the unrolled setting.** See Figure 12 (left) for the empirical growth trends of MAP estimates of the Unrolled DPG model ($D = 200$) parameters on $ER_{\frac{1}{4}}$ graphs. Parameter $\theta$ seems to follow the trend predicted by our analysis, but $\delta$ does not. Indeed it seems to follow a strongly linear trend, and $b$ seems to follow a power law. We can extrapolate from these trends to parameterize Unrolled DPG networks on significantly larger graphs with excellent performance; see Figure 12 (left).

**Larger ER graphs are smoother.** We also notice an interesting trend when constructing smooth graph as in Section 3. We find that the analytical distance matrices produced by $ER_{\frac{1}{4}}$ graphs get smoother as their size increases. For instance, see Figure 12 (right) where the mean and maximum entries of the analytic

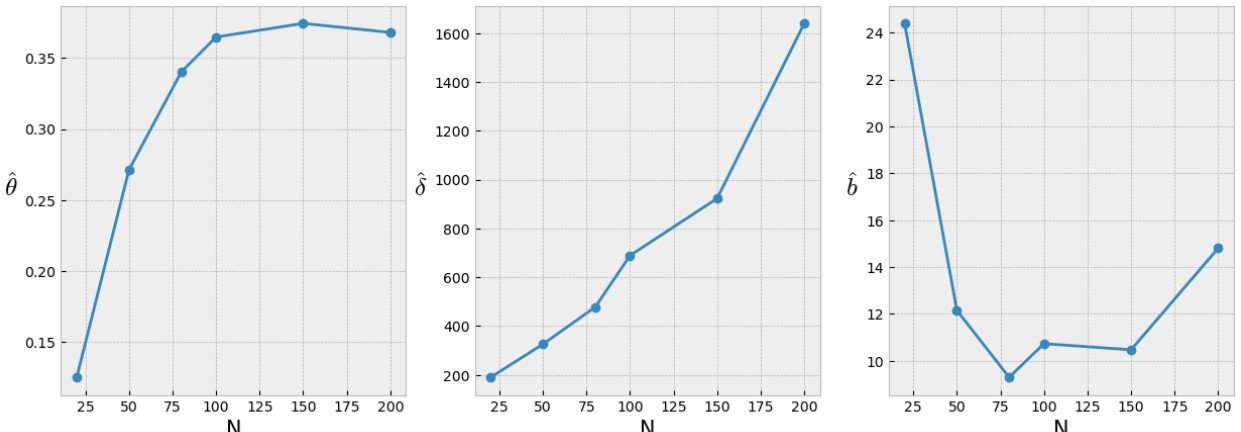

Figure 11: Learning MAP estimates $\hat{\Theta}$ of the DPG parameters on $\mathrm{ER}_{\frac{1}{4}}$ graphs of increasing size $N$ shows that $\hat{\theta}(N)$ resemble logarithmic growth, $\hat{\delta}(N)$ are linear, while $\hat{b}(N)$ show a sort of power law decay.

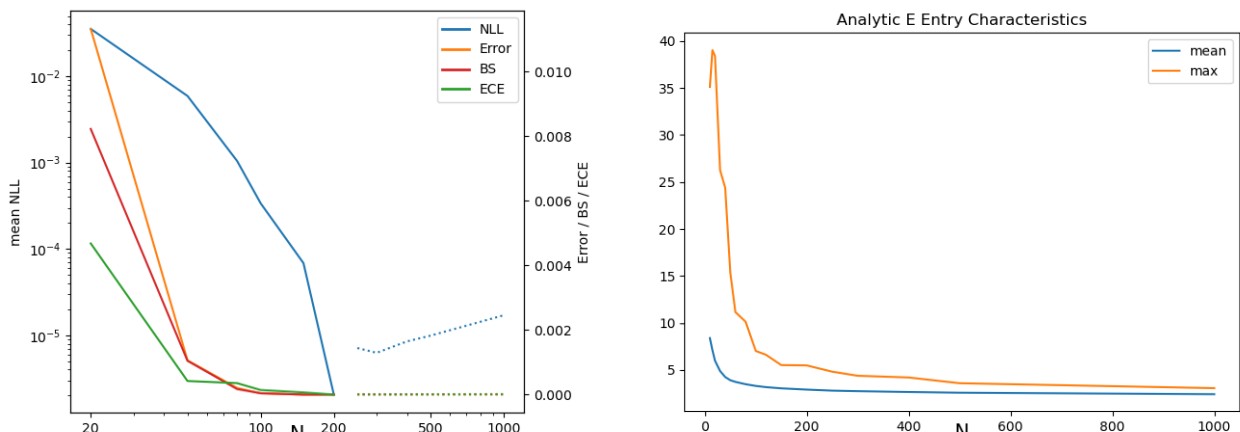

Figure 12: *Left*: Computing MAP estimates of DPG at increasing graph sizes shows improving recovery performance (solid lines). Extrapolating from the trends using the approach outlined in Appendix A.3, we can perform transfer learning larger graph sizes (dotted lines) with gradual decay in performance. *Right*: The analytic distance matrix E get smoother for increasingly sized $\mathrm{ER}_{\frac{1}{4}}$ graphs, providing an explanation for why graph recovery improves with larger graph sizes.

distance matrix get smaller as $N$ grows larger. This explains why the graph recovery problem gets easier and thus our performance gets better, as shown in Figure 12 (left).

### A.4 Further Experiments

**Prior modeling.** The prior modeling techniques introduced in Section 5 work across multiple random graph ensembles. Here, we replicate the prior modeling experiments done on $\mathrm{RG}_{\frac{1}{3}}$ in Section 5 for $\mathrm{ER}_{\frac{1}{4}}$, and $\mathrm{BA}_1$. See Figure 13 and Figure 14 and compare to Figure 3 (left).

**I.i.d. generalization across multiple random graph distributions.** Our method is performant across the random graph distributions we've tested. To supplement the results shown in Section 6, we show i.i.d. generalization results across more random graph distributions, as shown in Table 2.

**Data efficiency.** To further investigate how robust DPG is to sparse data settings, in Figure 15 we display i.i.d. generalization performance as a function of training set size $T$ on $\mathrm{RG}_{\frac{1}{3}}$ graphs and their corresponding

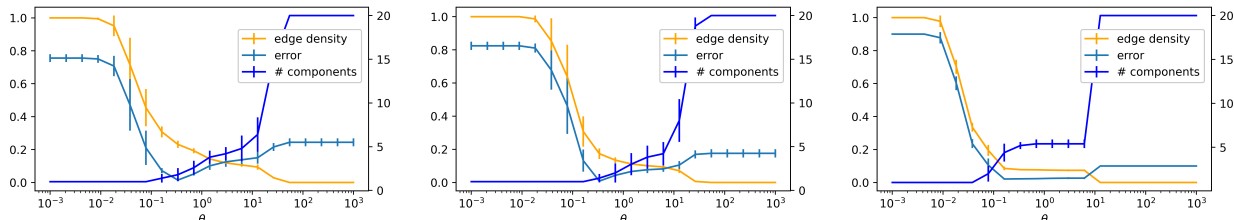

Figure 13: Prior modeling of $\theta$ for synthetic ensembles $\text{RG}_{\frac{1}{3}}$, $\text{ER}_{\frac{1}{4}}$, and $\text{BA}_1$ from left to right. The edge density and number of connected components do not require the presence of labels, while the error does. A threshold of $10^{-5}$ is used to decide the existence of an edge in order to remove the influence of small numerical effects.

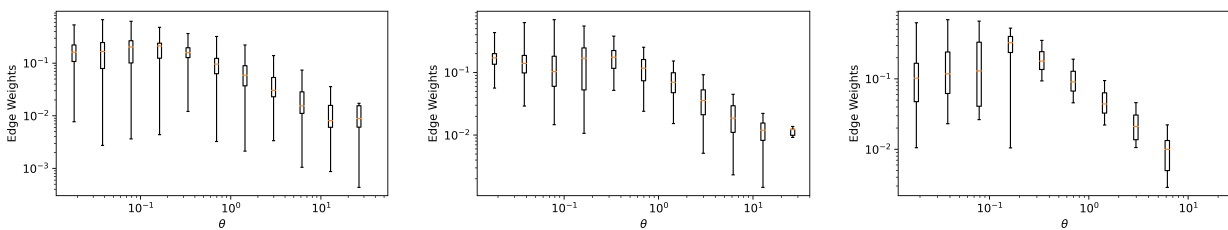

Figure 14: Prior modeling of $b$ and $\delta$ for synthetic ensembles $\text{RG}_{\frac{1}{3}}$, $\text{ER}_{\frac{1}{4}}$, and $\text{BA}_1$ from left to right. We inspect the edge weight distributions recovered running the held out data through Algorithm 1 for discretized values of $\theta$. The recovered edge weights are non-negative, and so the relevant scaling of $\delta$ and $b$ is a function of the median and maximum values.

analytic distance matrices. We additionally visualize the parameter posterior marginals. We find that past $T \approx 50$, i.i.d. generalization did not significantly improve, while past $T \approx 10^2$, the parameter posterior marginals converge.

**Prior ablation study**. In sparse data regimes DPG is more performant and has more efficient inference than DPG with un-informative priors (DPG-U). Indeed when the lack of data weakens the likelihood relative to the prior, a useful prior should maintain efficient Bayesian inference (Gelman et al., 2017). Table 3 illustrates this using $\text{RG}_{\frac{1}{3}}$ graphs and their corresponding analytic distance matrices on data sets of varying sizes $T$; see Section 3 for further discussion.

Table 2: I.I.D. Generalization of DPG across Random Graph Distributions

| Graph Distribution | NLL | BS ($\times 10^{-2}$) | Error (%) | ECE ($\times 10^{-3}$) |
|---|---|---|---|---|
| $\text{RG}_{\frac{1}{2}}$ | $8.993 \pm 9.516$ | $0.988 \pm 0.945$ | $1.111 \pm 1.267$ | 4.37 |
| $\text{BA}_1$ | $14.724 \pm 3.704$ | $2.614 \pm 0.751$ | $2.916 \pm 1.003$ | 30.63 |
| $\text{ER}_{.15}$ | $7.677 \pm 5.021$ | $0.997 \pm 0.569$ | $1.237 \pm 0.879$ | 3.04 |
| $\text{ER}_{\frac{1}{2}}$ | $0.962 \pm 3.029$ | $0.090 \pm 0.236$ | $0.084 \pm 0.265$ | 0.94 |

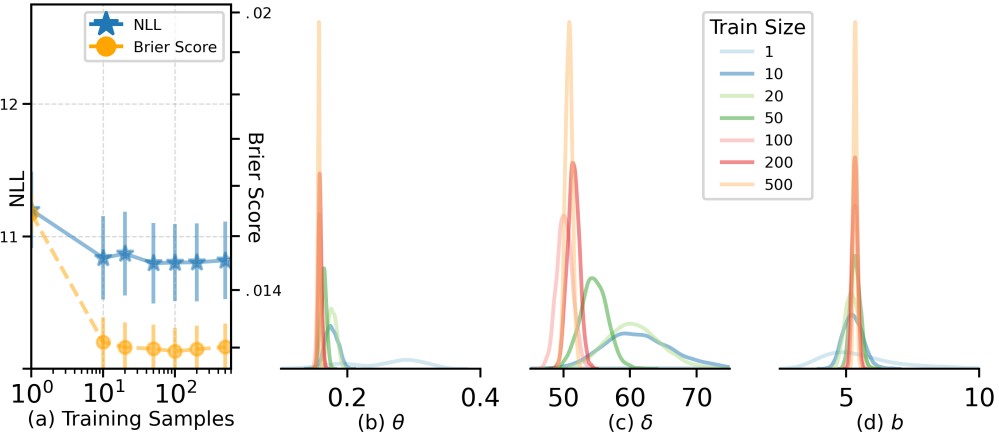

Figure 15: Impact of training set size on i.i.d. generalization performance. We use DPG with $D = 200$ on $\text{RG}_{\frac{1}{3}}$ graphs with analytic distance matrices to estimate generalization. The leftmost plot shows the generalization performance converge after $\approx 50$ samples on these same test graphs (error bars indicate $.05*\text{stdv}$ for compact visualization). The rightmost plots depict posterior marginals for the 3 parameters - $\theta$, $\delta$, and $b$ - for increasing sizes of training data. The observed reduction in variance of the posterior is a reflection of lower epistemic uncertainty. The posterior shows little change after $\approx 10^2$ samples.

Table 3: Ablation Study: Removing Informative Priors with Limited Training Data.

| | Model | Time (s) | NLL | BS ($\times 10^{-2}$) | Error (%) | ECE ($\times 10^{-3}$) | $\text{ESS}_\theta$ |
|---|---|---|---|---|---|---|---|
| $T = 50$ | DPG-U | 343.97 | 11.185 | 1.394 | 1.721 | 3.41 | 3.9 |
| | DPG | 44.15 | 11.183 | 1.392 | 1.721 | 3.32 | 19.7 |
| $T = 5$ | DPG-U | 15.15 | 11.476 | 1.508 | 1.847 | 5.62 | 15.5 |
| | DPG | 10.63 | 11.398 | 1.506 | 1.842 | 4.87 | 15.7 |
| $T = 2$ | DPG-U | 17.06 | 12.125 | 1.540 | 1.821 | 8.75 | 10.5 |
| | DPG | 10.66 | 11.661 | 1.522 | 1.832 | 7.55 | 11.3 |
| $T = 1$ | DPG-U | 11.11 | 11.942 | 1.673 | 1.789 | 7.98 | 7.1 |
| | DPG | 6.59 | 10.976 | 1.498 | 1.642 | 4.23 | 7.8 |

