# OpenReview forum: "Graph Structure Learning with Interpretable Bayesian Neural Networks"
_TMLR — Accepted by TMLR_

### Review · Reviewer_n2MU · 2024-07-07

**Summary Of Contributions:**

The paper introduces an approach to Graph Structure Learning (GSL) using Bayesian Neural Networks (BNNs) trained on smooth signal observations. Traditionally, GSL infers graph structures from nodal data via convex inverse problems or supervised deep networks. However, these methods often lack robust uncertainty quantification. The proposed method leverages independently interpretable parameters and Bayesian priors to enhance GSL's accuracy and reliability in uncertain data settings, such as medicine and finance.

**Audience:**

Yes

**Claims And Evidence:**

Yes

**Requested Changes:**

* Highlight the technical contribution in the context of related work.
* Introduce rigorous mathematical definitions of independently interpretable parameters.

**Strengths And Weaknesses:**

Strengths:
* The paper combines Bayesian principles with neural network architectures to address the GSL problem, offering a new paradigm for uncertainty-aware graph structure inference.
* Unlike traditional methods that require extensive tuning and lack interpretability in parameter settings, the introduced approach utilizes Bayesian learning to incorporate prior knowledge into the model's parameters, enhancing its applicability in domains where understanding the influence of specific parameters is crucial.
* The approach relies on standard regularizations on the graph loss and structure, facilitating the use of fast optimization algorithms. This ease of implementation makes the method accessible for large-scale problems.

Weakness:
* The technical contribution seems to be incremental. The main method is an adaption of (weighted) L1 regularization on the loss as well as the structure, i.e., eq (1) to (4). The dual proximal gradient descent solution algorithm is also pretty standard.
* A main contribution is the so-called independently interpretable parameters. However, the term "independence" does not strictly imply statistical independence as defined in probability theory. Instead, it refers to the interpretability of individual parameters in influencing specific characteristics of the inferred graph structure. In particular, the authors design the parameters to control certain graph properties, such as edge sparsity or connectivity patterns.

---

> ### Author Response · Authors · 2024-08-15
> **Authors’ response to Reviewer n2MU**
>
> Thanks for your time and effort in reviewing our manuscript, as well as for recognizing the novelty, applicability, and ease of implementation of this work. We appreciate your valuable suggestions to improve the paper’s presentation and its clarity. Point-by-point responses to your comments and associated requests for changes follow. We strive to improve our paper and will be happy to continue the discussion if any outstanding issues remain.
>
> &nbsp;
>
> **On the role of the DPG algorithm in our contributions**
>
> It is true that the inverse problem in (3) was originally formulated in (Kalofolias, 2016), and we give due credit in Section 3.1. On the other hand, the Dual Proximal Gradient Descent (DPG) iterations to solve (3) are a novel contribution of this work and offer several attractive features outlined in the second paragraph of Section 3.2. The DPG algorithm requires the fewest operations per iteration and the fewest number of parameters among existing solvers of (3), no step-size, and the discussed interpretability property w.r.t. graph sparsity. While admittedly DPG may not exhibit the faster convergence rate guarantees of the algorithms in (Saboksayr & Mateos, 2021) and (Wang et al., 2023), this is of secondary importance here because our goal is to *truncate*  and use the DPG iterations as a neural network (NN) blueprint by invoking the principle of algorithm unrolling (Section 4.1).
>
> These clarifications notwithstanding, we emphasize that the DPG iterations crucially facilitate our *primary technical contribution*: uncertainty estimation over edge predictions in the context of graph structure learning (GSL) from smooth signal observations. To our knowledge, this is the first work to address said problem. The DPG algorithm is ideally conducive to a true NN GSL unrolling, which in turn is uniquely compatible with Bayesian modeling (incorporation of prior information and predictive checks), and high fidelity inference methods (MCMC) as described in Sections 4 and 5. Crucially, the proposed Bayesian NN (BNN) offers high-quality and well-calibrated edge uncertainty estimates in GSL from smooth signals (evidence is provided in the experiments reported in Section 6).
>
> &nbsp;
>
> **Formalizing independent interpretability**
>
> Following your suggestion, we have introduced a formal definition of independent interpretability; please check Definition 1 in Section 3.1 of the revised manuscript. To close Section 3.1, we use Definition 1 to unequivocally  argue why $\theta$ is independently interpretable w.r.t. sparsity of the estimated graph adjacency matrix.
>
> Your point on the term *independent* possibly leading to confusion with the probabilistic/statistical notion is well taken. After including Definition 1, we believe there is no longer room for such ambiguity. But if you still feel strongly against the adopted terminology, we will be happy to instead use e.g., individually interpretable parameters.
>
> Thanks for this suggestion, which we believe contributed to adding clarity to the work.
>
> &nbsp;
>
> **Technical contributions in context**
>
> The main technical contributions in the context of related work are as follows:
> * Unlike prior optimization algorithms for GSL from smooth signals, which can be complex and contain nuisance parameters like step-sizes, we develop DPG which is highly compact and devoid of any such nuisance parameters. When unrolled, it yields the first true NN for GSL from smooth signals. This is to be contrasted, for instance, with the Unrolled PDS model in (Pu et al., 2021).
> * Unlike prior probabilistic approaches to GSL, we are the first to use BNNs for uncertainty quantification over edge predictions. To our knowledge, there are no other works in the literature that address the problem of uncertainty quantification in GSL from smooth signals.
> * Unlike Bayesian GNN approaches, we do not require partial observation of graph structure.
> * Other unrolling-inspired Bayesian deep networks have limited parameter interpretability and high dimensionality; see e.g., (Ekmecki & Cetin, 2022), which leads to naive priors and coarse posterior approximation. Instead, we introduce a concrete and actionable definition of parameter interpretability and leverage it for use in high-fidelity posterior approximation methods.
>
> Following your suggestion, we have updated the Related Works section to highlight the contributions in context; please check Section 2 in the revised manuscript. Under the **Summary of contributions** addendum that closes Section 1, we have also made explicit that bullet points therein are highlighting our technical contributions.
>
> &nbsp;
>
> Thanks again for your review.

---

### Review · Reviewer_yaD2 · 2024-08-06

**Summary Of Contributions:**

This paper proposes a method for graph structure learning (GSL) using Bayesian Neural Networks (BNNs), motivated by the importance of uncertainty estimates in many applications. The introduced method utilizes independently interpretable parameters in a novel iteration process, allowing for Bayesian modeling. Through using BNNs, the proposed approach is able to provide well-calibrated estimates of uncertainty on several benchmarks.

**Audience:**

Yes

**Broader Impact Concerns:**

No I don't have concerns on the ethical implications of the work

**Claims And Evidence:**

Yes

**Requested Changes:**

NA

**Strengths And Weaknesses:**

### Strengths

1. Uncertainty estimate is indeed important for many domains. This paper proposes the first BNN for supervised GSL that could naturally estimate uncertainty over edge predictions.
2. The proposed method is novel with sound, theoretical insights.
3. Good empirical results are reported on both synthetic and real data.

### Weaknesses

This paper is generally hard for me to follow, possibly because I am not very familiar with this direction and related works. That being said, I am not confident on my judgement of the paper.

---

> ### Author Response · Authors · 2024-08-15
> **Authors’ response to Reviewer yaD2**
>
> Thanks for your time and effort in reviewing our manuscript, as well as for finding our contributions novel, with sound theoretical insights and good empirical performance. We appreciate you recognizing the importance of uncertainty quantification in edge predictions and beyond.
>
> We strive to improve our paper and to convey our research findings in the most clear and accessible way to the broad TMLR readership. Please let us know if there is anything else that could help in your evaluation of this work. We will be happy to continue the discussion if any outstanding issues remain.
>
> Thanks again for your review.

---

### Review · Reviewer_rerB · 2024-08-06

**Summary Of Contributions:**

This work considers the problem of learning the structure of a graph from nodal observations alone in a supervised manner, i.e., when nodal observations come with corresponding graph labels. The prior literature has already formulated this problem as a convex optimization problem with a unique optimal solution. The challenge with the standard formulation though is that it depends on intertwined and non-interpretable parameters with respect to relevant graph characteristics, such as sparsity or edge weight magnitude. To deal with this, some prior work has introduced an alternative equivalent parameterization, which partially decouples the parameters. A critical insight of this work is that optimization can be performed on this alternative parameterization in an unrolled manner. In particular, the authors adapt the dual proximal gradient method (DPG) from prior works to the equivalent parameterization. However, the proposed DPG algorithm discards some uninterpretable parameters such as the step size, and requires fewer but interpretable parameters. By unrolling the DPG iterations, the authors argue that they obtain a true neural network (NN), which inherits a minimal number of interpretable parameters, while layers in unrolled DPG are simpler. By imposing prior distributions over the various parameters, we end up with a Bayesian NN (BNN) that additionally quantifies uncertainty. Because of the simpler unrolled network, it is possible to get a good posterior approximation via MCMC with uncertainty quantification. Furthermore, the Bayesian nature enables various tools, such as prior or posterior predictive checks. The authors conduct an extensive experimental study using both real and synthetic data, and show that DPG can produce good uncertainty estimates while outperforming other competitors.

**Audience:**

Yes

**Broader Impact Concerns:**

No concern.

**Claims And Evidence:**

Yes

**Requested Changes:**

I would be eager to raise my score since there are many positive things about this work, but I first need to clarify the scope with the authors.
- The authors should explain any symbols they have not introduced, such as $L^\dagger$ or $\lambda^\dagger_i$.
- It would help to clarify that the norm $||\cdot||_{1,1}$ is essentially a vector norm (or elementwise matrix norm) but not the $l_1$ matrix norm.
- Why do the authors not compare the proposed method to other methods from the GSL literature on smooth signals? I was thinking in particular about (Wang et al., 2023), which, according to its authors, significantly outperforms both Primal-Dual (Kalofolias, 2016) as well as FDPG (Saboksayr & Mateos, 2021). Wouldn't it make sense to compare the new method to the state-of-the-art algorithms in the GSL literature from smooth signals, i.e., to (Wang et al., 2023)?
- It indeed seems that this is the first work that uses a Bayesian neural network. But it certainly is not the first work using Bayesian methods for the purpose of graph structure learning. I would have expected to see either some detailed discussion or even experiments detailing how the proposed method does compared to other Bayesian methods on GSL. I believe this is quite important, otherwise the scope of the work is significantly reduced, and this work is only about the GSL literature from smooth signals (where, incidentally, the authors do not even compare to FDPG). I believe it is important for the authors to clearly define the scope of this work, and at least provide some discussion (if not experiments) on how their framework is related to other Bayesian methods for GSL. This will further strengthen the claims made by the authors, as everything will be seen under the right scope.
At the end of the day, the question is whether this work should be only seen as a Bayesian adaptation of FDPG, or as a viable framework for Bayesian GSL with the potential to outperform other Bayesian GSL methods.

**Strengths And Weaknesses:**

Strengths
- This is the first work using a BNN for graph structure learning with uncertainty quantification. The BNN is a true BNN, and contains only few interpretable parameters, which makes it possible to apply MCMC methods. It is also positive that the proposed method gets rid of the step-size parameter.
- The Bayesian treatment goes quite deep, and even includes aspects of prior modeling and (prior and posterior) predictive checking. This is very important in the context of uncertainty quantification.
- Using the iterative optimization as a neural network blueprint is a clever idea, especially in conjunction with the stochastic model.
- The experimental evaluation is quite extensive and contains both synthetic and real datasets. Furthermore, the authors provide various examples (backed with experimental evidence) in order to shed light on various theory aspects.

Weaknesses
- In the equivalent parameterization, $\theta$ is independently interpretable w.r.t. sparsity, but this is not the case for $\delta$. So, even though it is true that the equivalent parameterization is more interpretable than the original one, the two parameters are still intertwined.
- In (1), I think the authors are technically wrong to claim that the norm is the $l_1$-norm of matrix $\mathbf{Z}$. Indeed, the $l_1$-norm of a matrix is the maximum of the absolute column sums. What the authors instead mean is that $||\cdot||_{1,1}$ is the elementwise norm-1 of $\mathbf{Z}$, that is, the norm-1 of the vectorized form of $\mathbf{Z}$. Notice that (Kalofolias, 2016) explicitly mentions the elementwise character of that norm. This is not a serious error, but it is still good to be accurate and rigorous, in order to distinguish between vector norm and matrix norm.
- I think the authors introduced the symbol $L^\dagger$ without explaining what it means. They probably mean the Moore-Penrose pseudoinverse, but this should be stated clearly in the text. Furthermore, it was not clear what $\lambda^\dagger_i$ represents.
- In the empirical evaluation, the main competitor that the authors use is Unrolled PDS. It would be interesting to include more competitors . e.g., the algorithm in (Wang et al., 2023).
- Overall, in the experimental study the authors do not compare against other Bayesian methods for graph structure learning. Instead, they focus on the GSL literature from smooth signals. However, the topic of GSL with uncertainty quantification is quite big and more general, e.g., (Bayesian Structure Learning with Generative Flow Networks, Deleu et al., UAI 2022). To me, this work would have been much more convincing if the authors also compared the results that they get to the results of other Bayesian GSL frameworks.

---

> ### Author Response · Authors · 2024-08-15
> **Authors’ response to Reviewer rerB (Part 1)**
>
> Thanks for your time and effort in reviewing our manuscript, as well as for recognizing the novelty of our Bayesian Neural Network (BNN) approach and its attractive features, the depth of our Bayesian treatment, and our comprehensive validation protocol. We appreciate your suggestions to improve the paper, especially when it comes to delineating the scope of the problem being dealt with here. Point-by-point responses to your comments and associated requests for changes follow. We strive to improve our paper and will be happy to continue the discussion if any outstanding issues remain.
>
> &nbsp;
>
> **Undefined symbols**
>
> We appreciate your pointing out this lapse. Apologies for the lack of clarity. Indeed, we did not specify $\mathbf{L}^{\dagger}$ as the Moore-Penrose pseudoinverse, and did not define its associated eigenvalues $\lambda_i^{\dagger}$. This notation is now explained in the revised manuscript; please check the opening of Section 6.2. Likewise, immediately after (1) we have clarified that $\| \cdot \|_{1,1}$ stands for the matrix norm induced by the vector $\ell_1$-norm.
>
> &nbsp;
>
> **Graph structure learning (GSL) from smooth signals baselines**
>
> Indeed (Wang et al., 2023) is an impressive work, which we deservedly bring up in Section 3.2 as having the strongest convergence rate guarantees among *model-based* approaches for GSL from smooth signals. Similar comments apply for the fast dual-based proximal gradient (FDPG) algorithm in (Saboksayr & Mateos, 2021), but the convergence rate of FDPG is sublinear. Now, none of these model-based algorithms have been unrolled in the literature so they cannot be applied in the supervised learning setting dealt with here; see the opening of Section 4 for the formal problem statement. Moreover, none of them offer uncertainty quantification over edge predictions.
>
> What follows is our logic on the choice of the baseline methods selected for the experimental evaluation:
> * The primal-dual splitting (PDS) algorithm in (Kalofolias, 2016) is the *only* previous GSL method for smooth signals to be unrolled in (Pu et al., 2021), and so warrants inclusion. But even so, we faced significant difficulty in incorporating PDS into our Bayesian framework, mostly due to its step-size parameter, which causes instability; see the first paragraph of Section 6.1 and discussion in Appendix A.1.
> * The iterations introduced by (Wang et al., 2023) involve $2$ step-sizes (and $6$ total parameters), compounding the issues faced with PDS.
> * Attempts to unroll FDPG faced difficulty, a common observation for iterations with complex operations (here a result of its Nesterov-type acceleration); see e.g., (Monga et al., 2021). Without a reliable FDPG unrolling, Bayesian FDPG is untenable. This observation should reinforce our choice of DPG for subsequent unrolling and construction of the BNN model; see also our response to Reviewer n2MU.
>
> We reiterate that, to the best of our knowledge, there are no existing BNN approaches for GSL. Still, to strengthen our experimental evaluation we opted to implement a BNN with Unrolled PDS as its NN model, a method that has not been published elsewhere. So we have gone over and above to include the most meaningful baselines and offer solid evidence supporting the claims made in the paper. For the reasons elaborated above, we respectfully believe it is not feasible (and arguably meaningful) to run comparisons with the model-based algorithms in (Wang et al., 2023) and (Saboksayr & Mateos, 2021). We sincerely hope this justified omission does not diminish from appreciating DPGs benefits, namely its interpretability, simplicity, parameter/runtime efficiency, and resulting compatibility with the Bayesian modeling process.
>
> Clarifying comments along these lines have been included at the end of the first paragraph in Section 6.1.
>
> &nbsp;
>
> **Scope and other Bayesian methods for GSL**
>
> As we state in the Abstract, and subsequent main body sections including the Introduction, Related Work, and the problem statement in the openings of Sections 3 and 4, the *scope of this work* pertains to undirected GSL from smooth signals in a supervised setting. We develop a novel BNN to tackle this problem, which produces a distribution over unseen test graphs allowing estimation of uncertainty over edge predictions. One should appreciate that the Bayesian modeling methodology discussed is general, meaning what we present is a BNN template which uses a NN produced by unrolling an optimization algorithm to solve some convex inverse problem with independent interpretability. However, our focus is exclusively on GSL from smooth signal observations – a problem that is fairly broad and important in its own right.

---

> > ### Author Response · Authors · 2024-08-15
> > **Authors’ response to Reviewer rerB (Part 2)**
> >
> > We agree that there have been Bayesian methods adopted to tackle GSL problems (some of these are discussed in the second paragraph of Section 2 - Related Work), but GSL instances come in many shapes and forms and it is not straightforward to carry out comparisons between any two methods of this fairly broad class. For instance, a tomographic network topology inference problem is markedly different from a link prediction task; see e.g. Ch. 7 in (Kolaczyk, 2009). Both of these are GSL instances, but they substantially differ from the GSL problem dealt with here (see Section 3).
> >
> > We appreciate you bringing up the nice work in (Deleu et al., 2022), which we now cite and briefly discuss in Section 2 of the revised manuscript. But this only reinforces our point, since the generative flow approach therein aims to learn *directed* acyclic graphs underlying probabilistic graphical models known as Bayesian networks, often of varying nodal sizes, while we address inference of *undirected* graphs from smooth signal observations. The Bayesian paradigm is certainly a common theme, but we hope we can agree that these two works are fundamentally different in their scope and general objectives.
> >
> > &nbsp;
> >
> > Thanks again for your review.

---

> > ### Comment · Reviewer_rerB · 2024-08-17
> >
> > I thank the authors for their responses. Some comments:
> > - I completely disagree that the norm under consideration is the matrix norm induced by the vector $l_1$ norm. Indeed, it is well know that this  operator norm  is given by the maximum  $l_1$-norm of all matrix columns. What the authors instead want is the sum of the absolute values of all matrix entries. This is precisely what the entry-wise $l_{1.1}$ matrix norm is about: it returns the $l_1$ norm of the vectorized matrix. If this is confusing to the authors, I suggest they read carefully various online resources/articles on the matrix norm, and the difference between matrix norms induced by vector norms vs. entry-wise or vectorized norms. It would be a pity if the submission contained such simple-to-fix errors.
> > - I appreciate the clarifications that the authors provided on GSL and the various other baselines. Regarding (Wang et al., 2023), what was not clear to me was whether the baseline without unrolling (i.e., the original algorithm by (Wang et al., 2023) was not included because it solves a different problem, and is thus not applicable for the supervised setting under investigation in this work. If that is the case, then it makes perfect sense why the authors would not include the original algorithm. If, on the other hand, it can be used for the tasks at hand (albeit without uncertainty quantification), then it could have been included as a competitor, even without unrolling. I believe this is what the authors meant, i.e., that the original algorithm by (Wang et al., 2023) cannot be used for the supervised setting under consideration without unrolling. But I could be wrong.
> > - I appreciate the authors' clarification on the paper scope.
> > - As the authors also pointed out, their Bayesian treatment and methodological approach are meticulous.
> >
> > I would appreciate it if the authors could provide their feedback on the first two points.

---

> > > ### Author Response · Authors · 2024-08-17
> > > **Thanks for the follow up**
> > >
> > > Thanks for taking the time to go over our responses and provide additional valuable feedback. Responses to your outstanding comments follow:
> > >
> > > - Your point on the matrix norm is well taken. We have revised the manuscript again to clearly state that $||\mathbf{Z}||_{1,1}$ denotes the *entrywise* $\ell_1$-norm of matrix $\mathbf{Z}$; please check the text immediately following (1) in the revised manuscript.
> > > - With regards to the second point you raise, your first appreciation is the correct one. The algorithm in (Wang et al., 2023) is not directly applicable to the supervised setting dealt with here. The same e.g., is true for the primal-dual splitting (PDS) algorithm used in (Kalofolias, 2016). In the paper we refer to these as *model-based* approaches to (point) estimation of graphs from smooth signals; there is no training set, learning, and generalization to unseen test data involved in those works. And as we have made it abundantly clear, there is no uncertainty quantification either. Unlike the solver in (Wang et al., 2023), (Pu et al., 2021) unrolled the PDS iterations to obtain a GSL neural network that can be trained from a set of nodal signals-graph label examples. This is the supervised GSL setting considered here as well, and so we indeed compare with Unrolled PDS in (Pu et al., 2021).
> > >
> > > Thanks again for engaging in the discussion and for your willingness to clarify important aspects of our work.

---

> > > > ### Comment · Reviewer_rerB · 2024-08-17
> > > >
> > > > I thank the authors for correcting the small error, but also for providing important clarifications on the applicability of other competitors. With all this in mind, I believe that the claims made by the paper are well supported. I have updated my review to reflect this.

---

### Comment · Action_Editor_Vz5D · 2024-08-21
**The final recommendation**

Dear reviewers,
thank you so much for your work, it is now time to submit your recommendation for this paper.
I kindly invite you to have one more lok at the following link
https://jmlr.org/tmlr/reviewer-guide.html
Looking forward to receive your recommendation.
All the best

---

### Decision · Action_Editor_Vz5D · 2024-09-03

**Recommendation:** Accept as is

**Comment:**

The concerns raised by the reviewers were all addressed in a satisfactory manner as also agreed upon by the reviewers raising the issues (rerB).

**Audience:**

All the reviewers agree that the manuscript fits well the interest for the machine learning research community.

**Claims And Evidence:**

According to all three reviewers claims from the manuscript are followed by sufficient evidence to support them.